# Improving public support for climate action through multilateralism

Michael M. Bechtel[1,2] ✉, Kenneth F. Scheve [3] ✉ & Elisabeth van Lieshout[4]

For decades, policymakers have been attempting to negotiate multilateral climate agreements. One of the motivations for securing cooperation among multiple states is the belief that the public will be more supportive of adopting costly climate policies if other countries do so, both because this makes it more likely that important sustainability goals will be reached and because those efforts resonate with widely held fairness norms. However, some recent research suggests that public approval of climate action is independent of the policy choices made by other countries. Here, we present two different experimental studies fielded in multiple countries showing that multilateralism significantly increases public approval of costly climate action. Multilateralism makes climate policy more appealing by improving effectiveness beliefs and the policy's perceived fairness. Pursuing climate action within a multilateral setting does not only promise improved policy impacts, but may also generate higher levels of public support. Preregistration: This study has been pre-registered at AEA RCT Registry under #AEARCTR-0004090.

Millions of people around the world have mobilized to protest the failure of governments to implement policies that would substantially reduce greenhouse gas emissions and address the climate challenge. It would seem that these concerns resonate not just with activists, but the majority of voters in many of the major emitter countries. Nonetheless, often when governments have acted to reduce emissions, they have been met by intense opposition, some of it certainly by organized special interest groups but some of it by voters reluctant to incur the costs of higher carbon prices.

How can we explain political opposition to costly climate action? One part of the answer is surely the scale of the distributional conflict associated with the sort of energy transition necessary to address the climate challenge. The costs and benefits of climate change and climate policy vary across individuals, generations, firms, industries, regions, and countries[1–5]. Even in surveys that initially seem to indicate that large majorities favor climate action, making the costs and their distribution explicit can quickly erode support for climate policy. Another common explanation points to the global public goods character of addressing climate change[6–11]. If sustainable greenhouse

gas emissions are a global public good with the usual characteristics, this may lead to climate policy underprovision.

Policymakers and researchers have invested significantly in identifying international agreements and institutions that would address the international component of the problem, e.g., through central cooperative frameworks such as the United Nations Framework Convention on Climate Change (UNFCCC). This practical and scholarly effort builds on insights into how small and large communities across diverse issue domains solve public goods problems[12–14]. A possible motivation behind policymakers' interests in creating a workable international framework for cooperation on climate change could be to secure greater public support for policies to reduce greenhouse gas emissions, especially policy measures that are costly to their voter base. We argue that voters value the benefits that climate action provide and are more willing to contribute to the global public good of manageable emissions if other countries are doing so as well because of effectiveness beliefs and reciprocity norms[15,16]: First, multilateral policies are more likely to be effective, i.e., able to realize important social, economic, and environmental sustainability goals. Second,

[1]Department of Political Science and European Affairs, Cologne Center for Comparative Politics, University of Cologne, 80870 Cologne, Germany. [2]Swiss Institute for International Economics and Applied Economic Research, CH-9000 St.Gallen, Switzerland. [3]Department of Political Science, Yale University, New Haven, CT 06520-8301, USA. [4]Department of Political Science, Stanford University, Stanford, CA 94305-6044, USA. ✉e-mail: m.bechtel@uni-koeln.de; kenneth.scheve@yale.edu

multilateral agreements resonate with reciprocal fairness norms which increase the willingness to bear the costs of climate action if other countries are exerting greater policy efforts to reach the collective goal of limiting global warming.

Early studies offer evidence that seems explicitly or implicitly consistent with the idea that the public values multilateral cooperation on climate change[16,17]. However, several more recent contributions make explicit claims about the limited relevance or irrelevance of the behavior of other countries for our understanding of the mass politics of climate change[18–20]. Instead, these studies suggest that public support for costly climate policies does not meaningfully depend on whether other countries are also contributing or not. We present evidence from two experimental studies in four countries—France, Germany, the United Kingdom, and the United States—indicating that voters do care about the policies of other countries and are more likely to support costly policies when other countries are also doing so. Moreover, we offer experimental evidence suggesting that this interdependence is due to how multilateralism—even if it merely revolves around loose policy coordination in which states make nationally determined contributions consistent with the flexibility offered by the UNFCCC framework—shapes the expected effectiveness of policies and how it resonates with reciprocal fairness norms.

In this work, we investigate whether multilateralism increases support for costly climate policies. Multilateralism may be thought of as describing a continuum of behavior in international politics. This continuum may range from decentralized cooperation among states, e.g., a set of two or more countries independently contributing to a public good in an ad-hoc fashion[21], to highly institutionalized forms of interaction in and through international organizations and agreements[22,23]. We study multilateralism in the context of climate cooperation where countries engage in "decentralized policy coordination"[24] which typically entails declaratory statements and agreements that specify policy objectives (e.g., the goal to limit global warming to well below 2 °C as set out in the Paris agreement) and possible means, but lack enforcement mechanisms[25].

### The causal effect of multilateralism on climate policy support

To assess whether multilateralism causes higher levels of policy support, we start by employing a vignette survey experiment in France, Germany, and the United Kingdom. We conduct the experiment as a module in original surveys fielded in April 2019. For each country, the samples are representative of the adult population with 2000 respondents in each. The Methods section provides a detailed description of the sampling frame. Supplementary Table 1 offers a comparison of the distribution of sociodemographic characteristics in the target population, the raw sample, and the weighted sample. All our results employ survey weights, but the findings are very similar when analyzing the unweighted data.

We investigate the effect of multilateralism on support for climate policy by exploring public support for a domestic carbon tax. This policy instrument is relatively easy for respondents to understand and it is reasonable to expect that it reduces carbon emissions. Critically, it can be implemented by either a single country or multiple countries consistent with the flexibility offered by the UNFCCC and nationally determined contributions to reducing global emissions. This means that our approach does not study the more institutionalized and specific approach of creating "linked carbon taxes" although the modifications needed to investigate this policy instrument in future work are straighforward. The exact wording of the vignette experiment is (with the placeholder COUNTRY being a respondent's country):

"Suppose COUNTRY [decides OR and other major economies decide] to implement a carbon tax, which is an additional tax on

the $CO_2$ content of fuels, to address climate change. Generally speaking, do you approve or disapprove of COUNTRY implementing such policies?"

We randomize whether a respondent receives a version of this question in which the carbon tax would be implemented unilaterally or whether it would be part of a multilateral setting in which other major economies also introduce a carbon tax. We record responses on a 1–10 (strongly approve - strongly disapprove) answer scale.

Higher levels of international participation in climate action may increase public support because of heightened expectations about the effectiveness of these policy efforts. We investigate the effectiveness mechanism by crossing the multilateralism experiment with a second vignette experiment that provides information about expected policy impacts. This allows us to analyze the causal mechanism in an eliminated effects framework[26]. The experiment consists of one control group—which receives no additional information—and two treatment groups *Effectiveness: Low* and *Effectiveness: High*. We provide these groups with the following additional primes:

*Effectiveness: Low*: "Most experts think this will avoid a few of the economically and environmentally damaging consequences of climate change."

*Effectiveness: High*: "Most experts think this will avoid most of the economically and environmentally damaging consequences of climate change."

## Results

We analyze the proportion of individuals approving the introduction of a carbon tax (levels of support that exceed 5, the midpoint of the scale). Figure 1a reports carbon tax support by the randomly assigned multilateralism condition along with 95% confidence intervals. In the control condition that did not receive any effectiveness information, we find that about 53% of all respondents support the introduction of a carbon tax if this policy is pursued unilaterally. It is noteworthy that this estimate is not different from 50% in statistical terms. Therefore, in the absence of multilateral policy coordination, we would not be certain that a majority actually supports the introduction of a carbon tax. However, when other countries are merely mentioned to also introduce a carbon tax, support is 59%, which is equivalent to a 6 percentage points increase over the unilateralism condition. We also note that this treatment effect is equivalent to an 11% increase over the baseline level of unilateral carbon tax approval. Given the relative simplicity of the treatment, this is a substantively important marginal effect. Moreover, a multilateral approach generates a level of support that significantly exceeds 50% in statistical terms, thereby contributing to a more certain majority in favor of introducing a carbon tax. These results are very similar when analyzing the unweighted data (Supplementary Fig. 1) but differ somewhat by country (Supplementary Fig. 2).

We are interested in exploring effectiveness as a potential mechanism capable of explaining the appeal of multilateralism. We perform this investigation in an eliminated effects framework. First, recall that our estimate of the average treatment effect above is the difference between the level of carbon tax support in the unilateral condition and the carbon tax approval in the multilateralism condition in which respondents did not receive information about effectiveness. Second, the estimate of the effect of multilateralism for respondents exposed to the low or high effectiveness condition is our estimate of the average controlled direct effect which is the part of the total effect that is not due to mediation by or interaction with effectiveness. Third, the average treatment effect minus the average controlled direct effect is the eliminated effect[26]. This quantity is the portion of the average treatment effect that can be explained by the effect of the treatment (multilateralism) through the mediator

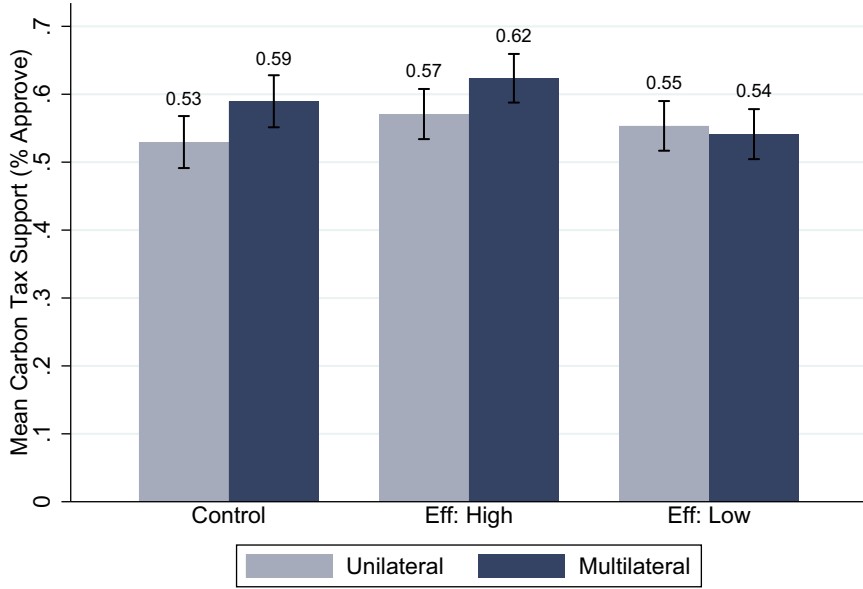

(a) Proportion Supporting a Carbon Tax by Multilateralism and Effectiveness Treatment Condition

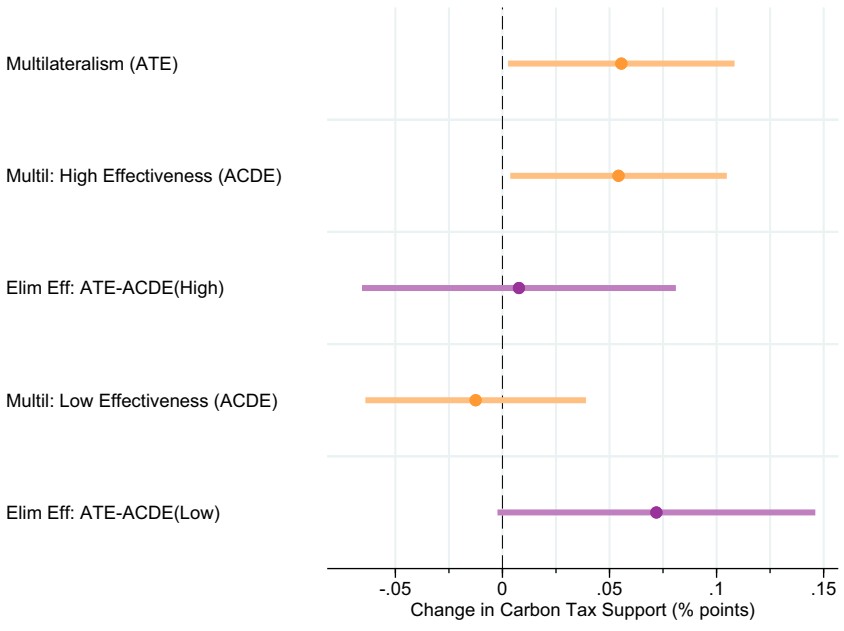

(b) Causal and Eliminated Effects (EE) of Multilateralism by Effectiveness Prime

**Fig. 1 | The effect of multilateralism on carbon tax support and the effectiveness mechanism ($N$ = 6000). a** The figure shows the proportion of individuals supporting the introduction of a domestic carbon tax by randomly assigned multilateralism and effectiveness conditions. **b** The figure reports the average treatment effect (ATE), average controlled direct effects (ACDE), and eliminated effects (EE) of multilateralism by randomly assigned effectiveness condition (High = high effectiveness, Low = low effectiveness) estimated using a linear probability model with robust standard errors. Regressions control for gender, age, income, education, and employment status. Country fixed effects included. Error bars indicate 95% confidence intervals. $N$ (France) = 2000, $N$ (Germany) = 2000, $N$ (United Kingdom) = 2000. Unweighted results are very similar, see Supplementary Fig. 1.

(effectiveness) and any interaction between multilateralism and effectiveness.

Figure 1a shows the results for the low and high effectiveness conditions in which we fix respondents' beliefs about policy effectiveness. We find that in the high effectiveness condition a unilateral carbon tax approach is backed by about 57% and that this proportion increases to 62% in the multilateralism treatment. In contrast, when policy effectiveness is low, the switch from a unilateral to a multilateral climate policy framework has almost no effect on carbon tax approval.

Figure 1b shows that multilateralism causes a significant increase in carbon tax support in the control and in the high effectiveness condition, but not in the low effectiveness condition. This means our two eliminated effect estimates provide contrasting evidence on how well effectiveness explains the impact of multilateralism on public support. For high effectiveness, the eliminated effect is very close to zero while for low effectiveness, it is positive at about 6 percentage points and close to being significant at the 5%-level. The latter estimate indicates that most of the effect of multilateralism is explained by effectiveness, but of course the caveat is

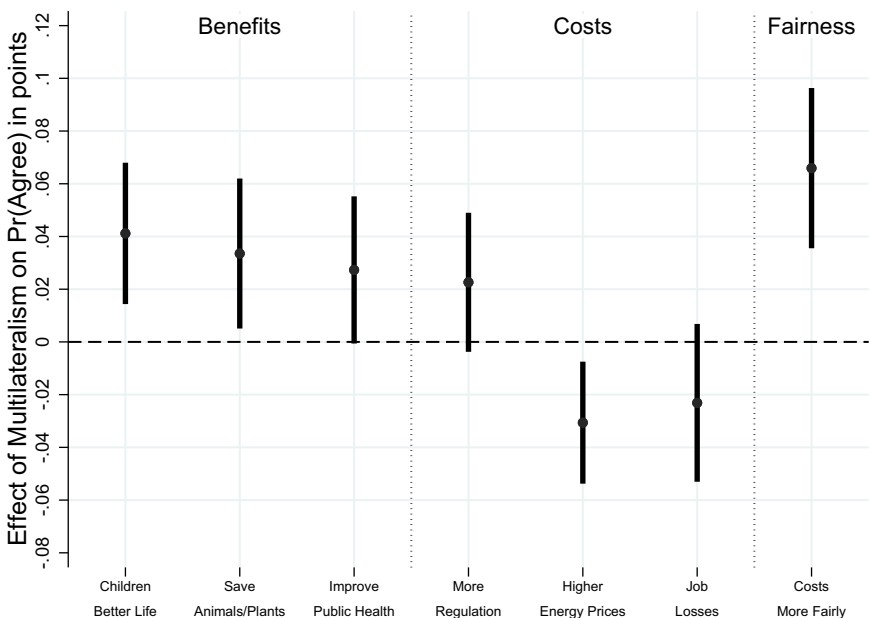

**Fig. 2 | The causal effects of multilateralism on sustainability beliefs: benefits, costs, and fairness of climate action in France, Germany, and the United Kingdom (*N* = 6000).** This plot reports coefficients from linear regressions of statement approval on a binary indicator that is one if climate action is multilateral and is zero if climate action is unilateral. Error bars indicate 95% confidence intervals. All regressions control for gender, age, income, education, and employment status. Country-fixed effects included. Survey weights applied. Results for the unweighted data are very similar, see Supplementary Fig. 3. *N* (France) = 2000, *N* (Germany) = 2000, *N* (United Kingdom) = 2000.

that the high effectiveness estimate is inconsistent with this interpretation.

## Multilateralism and expectations about effectiveness and sustainability

To explore which impacts individuals expect from a multilateral as opposed to a unilateral climate policy framework, we added a question after the multilateralism vignette experiment that prompted respondents to indicate whether they thought specific statements about a wide range of sustainability impacts were true or false. The statements covered environmental, economic, and fairness-related impacts of climate action, see Methods section.

We estimate how multilateralism changes effectiveness beliefs by regressing whether a statement is selected as true on a multilateralism treatment indicator, a full set of sociodemographic control variables, and country-fixed effects. The results in Fig. 2 indicate that multilateralism matters: individuals are significantly more likely to expect greater environmental benefits, lower economic and governance costs, and fairer cost distributions from multilateral efforts (see Supplementary Fig. 3 for the unweighted results which are quite similar). Regarding benefits, we find that multilateralism significantly improves beliefs about the ability to reach important sustainability goals, for example, the potential for climate action to improve the lives of respondents' children and grandchildren.

We find broadly similar effects when analyzing whether the policy will save endangered animals and plants and whether it will improve public health outcomes. When assessing the impact of multilateralism on costs, we find that a multilateral policy approach does not systematically affect concerns related to increased regulation and potential job losses caused by progressive climate action. However, multilateral initiatives reduce public unease about energy price increases[27]. Finally, multilateralism significantly increases the belief that the costs of climate action will be distributed more fairly, which is illustrative of the close linkages between climate policy and justice. Supplementary Fig. 4 reports the results by country, suggesting that the multilateralism effects are strongest in France and less systematic in Germany and the United Kingdom.

## Multilateral policy design and climate support

So far, we have explored the impact of multilateralism at the general level on public support for costly climate action. Our second study begins to unpack the notion of multilateralism by investigating whether individuals have preferences over the specific, multidimensional policies in other countries and whether those policies influence the willingness of respondents to adopt costly policies in their own country. This study was conducted as a separate module in the French, German, and British surveys described above and in an additional survey fielded in the United States in December 2018.

We constructed a randomized conjoint experiment that presented respondents with two multilateral climate policy scenarios and asked them to indicate which of the two they prefer. Each scenario specified a multilateral climate policy setting: a combination of costly climate policy decisions for both a respondent's own country and other major economies (Supplementary Fig. 5 shows detailed conjoint instructions along with an example profile). The policy features included average costs to households, their temporal distribution, and whether revenues would be invested in adaptation or mitigation. Cost levels are important because they are indicative of how much an individual or country is willing to contribute to reducing greenhouse gas emissions by increasing the costs of carbon (the possible values employed were monthly household costs approximately equal to 0.5%, 1%, 2%, or 2.5% of GDP). The cost schedule denotes the sequencing of carbon pricing over time, which we allowed to be held constant, increased over time, or decreased[28]. Investment mix refers to the share of resources spent on the two fundamental policy responses to climate change: adaptation and mitigation[29]. This distinction is important theoretically because previous studies have argued that adaptation efforts provide local benefits whereas mitigation contributes to the global public good of reducing greenhouse gas emissions which provides non-excludable and non-rival benefits. An important innovation of this conjoint for purposes of this study is that each policy dimension and its values were specified for both the respondent's own country and other major economies. This allows us to investigate the sensitivity of individuals to the policies of other countries and benchmark these

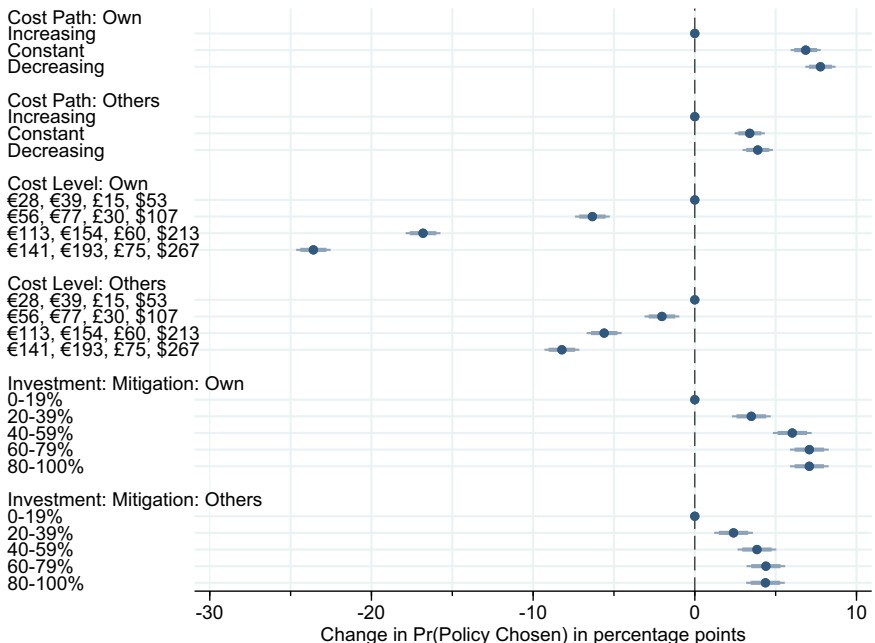

**Fig. 3 | The causal effects of climate policy choices by other countries on public support in France, Germany, United Kingdom, and the United States (N = 129,280).** This plot reports coefficients from linear regressions of policy approval on randomly assigned climate policy choices made by other countries. Error bars indicate 95 and 99% confidence intervals; points without lines indicate reference categories. Results for the unweighted data are very similar (see Supplementary Fig. 6) and quite stable across countries (see Supplementary Fig. 7). Results for each country separately are shown in Supplementary Fig. 6. N (France) = 32,000, N (Germany) = 32,000, N (United Kingdom) = 32,000, N (United States) = 33,280.

against the causal effects of domestic policy features. Each respondent completed eight conjoint choice tasks.

We estimate whether respondents care about the climate policy decisions made by other countries by regressing whether a policy was chosen on indicators for each of the fully randomized attributes. Figure 3 reports the results. We find that increasing the costs of carbon domestically has a strong impact on support for climate action: public support drops by 6 percentage points if costs increase from low €28 in France, €39 in Germany, £15 in the UK, and $53 in the US to medium levels (€56, €77, £30, $107) and declines by 17 percentage points if domestic costs are high (€113, €154, £60, $213). We find a weaker, but significant aversion to climate costs in other countries with effects sizes about one-third of those for domestic climate costs.

While the sensitivity of carbon tax approval to costs in other countries is consistent with the main claim of the paper that voters care about the policies pursued in foreign countries, they are also noteworthy from a public goods perspective. Classic models of free-riding would predict that individuals prefer foreign countries to make higher contributions since many of the benefits of climate action are non-excludable. Our results do not support this prediction. Instead, we believe that the aversion to increasing carbon prices in other countries is consistent with two potential explanations. First, costly climate action by other countries generates pressure on countries to reciprocate. Reciprocating, however, requires incurring costs and publics are averse to costs. A second potential explanation is that individuals hold other-regarding preferences such as altruism or inequality aversion. According to this view, publics dislike other countries to increase climate costs because this may hurt the financial well-being of individuals in other countries and add to rising inequality.

Turning to multilateral cost paths, publics generally prefer both their own and other countries to adopt constant or decreasing cost paths over the baseline of increasing cost schedules. Lastly, we also find that higher mitigation investments by other countries significantly increase support for multilateral climate policy. The effect is again somewhat smaller than the sensitivity to climate investment decisions in one's own country, but still sizeable: policy

support increases by about 5 percentage points if other countries invest at least 60% in mitigation efforts. This effect would be sufficient to offset the drop in public support due to increasing other cost levels from low to medium. A preference for mitigation investment versus adaptation in other countries is consistent with a concern about whether policy efforts will be effective as respondents in a given country are likely to benefit more from the mitigation efforts in other countries than from adaptation investments in those countries.

Our analysis so far has focused on providing evidence that individuals care about the climate policies of other countries. We now turn to the reciprocity-based question of whether these policies also influence the willingness of respondents to back costly policies in their own country. In addition to a concern about effectiveness, norms of reciprocity have been shown to support cooperation across diverse settings. One motivation for countries to seek multilateral agreements is to activate reciprocity norms that would increase the willingness of citizens to adopt costly domestic climate policies which would contribute to the global public good of reduced emissions.

We can use the conjoint data to investigate the role of two notions of conditional cooperation: qualitative and exact reciprocity. First, we explore qualitative reciprocity, i.e., whether the aversion to domestic costs depends on the climate efforts made by other countries. Specifically, reciprocity norms suggest that more costly climate action by other countries should lessen the distaste for costs in one's own country. We evaluate this predicted cross-country interdependence between the climate contributions made in foreign countries and domestic cost aversion by re-estimating the causal effects of own household costs on climate policy support for each of the four cost levels in other countries. Figure 4 shows the results.

We find that when other costs are set very low, a domestic policy that increases costs to medium levels reduces support by about 7 percentage points. In contrast, this effect shrinks to about 5.5 percentage points when other costs are very high. This pattern of a decreasing aversion to own costs when other countries are willing to adopt progressive climate pricing policies becomes more pronounced

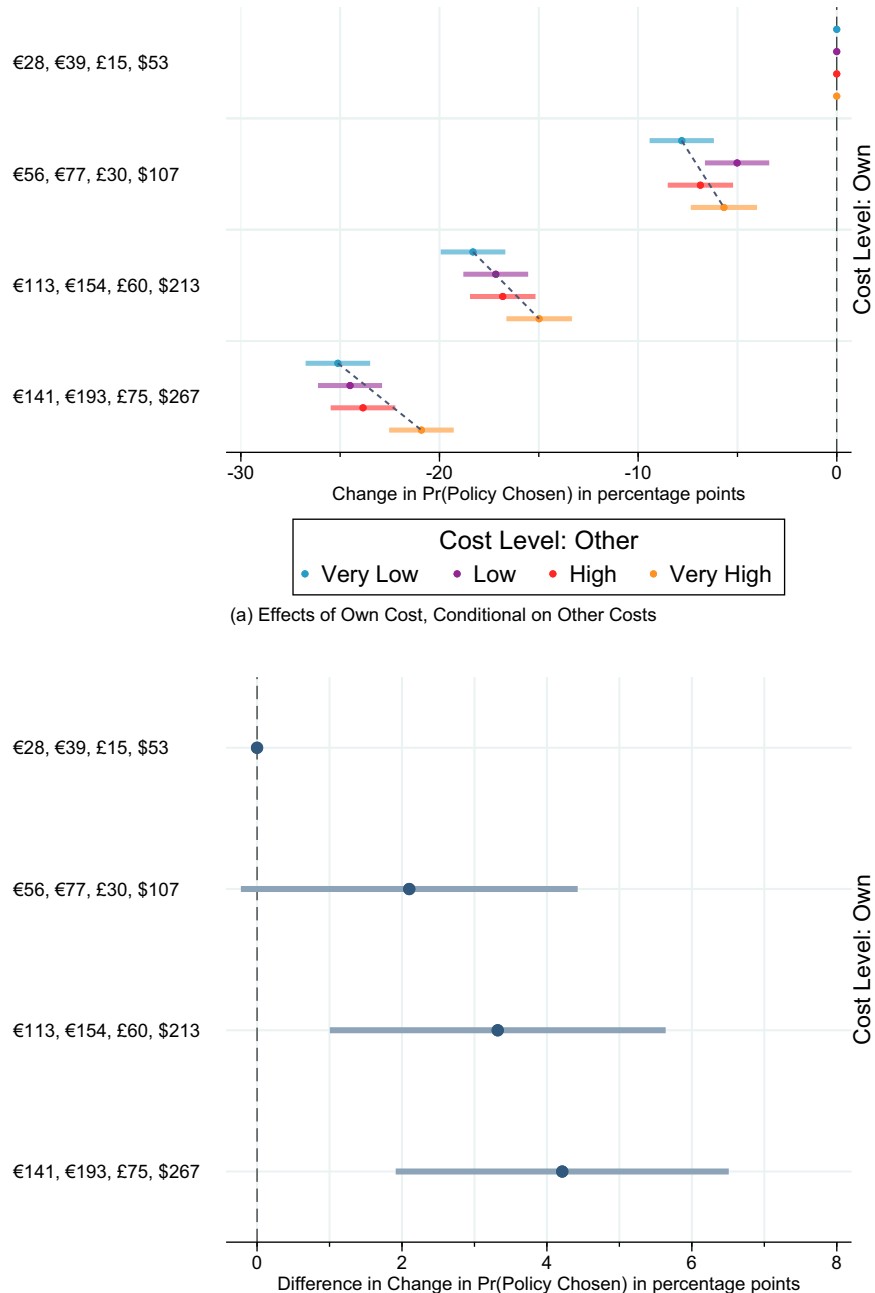

(a) Effects of Own Cost, Conditional on Other Costs

(b) Difference in Effect of Own Cost if Other Costs Increase from Very Low to Very High

**Fig. 4 | The causal effects of climate policy costs by other costs in France, Germany, United Kingdom, and the United States ($N$ = 129,280). a, b** The figure reports coefficients from linear regressions of policy approval on randomly assigned climate policy household costs. **a** The figure shows the effects of own cost conditional on other costs. **b** The figure reports the differences in the effects of own cost if other costs very high and own costs if other costs very low. Error bars indicate 95% confidence intervals. Results for the unweighted data are very similar, see Supplementary Fig. 8. Results by country are shown in Supplementary Fig. 9. $N$ (France) = 32,000, $N$ (Germany) = 32,000, $N$ (United Kingdom) = 32,000, $N$ (United States) = 33,280.

and statistically significant when own costs are high or very high. In the latter scenario, the causal effect of very high own costs drops from 25 to 21 percentage points on average. This finding is consistent with the reciprocity-based argument that individuals value the climate policy contributions made by other countries and this lowers their aversion to incurring high costs. The interdependence between domestic and foreign carbon pricing is all the more noteworthy since it has proven challenging to detect theoretically meaningful interactions between attribute features in conjoint data[30].

We can think of this analysis as showing evidence of qualitative reciprocity: individuals appear to be more willing to incur costs to

reduce emissions if other countries are making costly efforts. Our data also allow us to explore evidence for exact reciprocity understood as the increased willingness to support adopting climate policies domestically if these match the efforts of other countries. Exact matching seems a natural focal point for reciprocal behavior.

We test this expectation by investigating whether scenarios in which other countries match the cost levels of one's own country increases support for climate action more strongly than non-matching contributions. We compute the average marginal interaction effects (AMIEs) as defined by ref. 31. The AMIE quantity of interest captures the additional effect on the probability that a policy

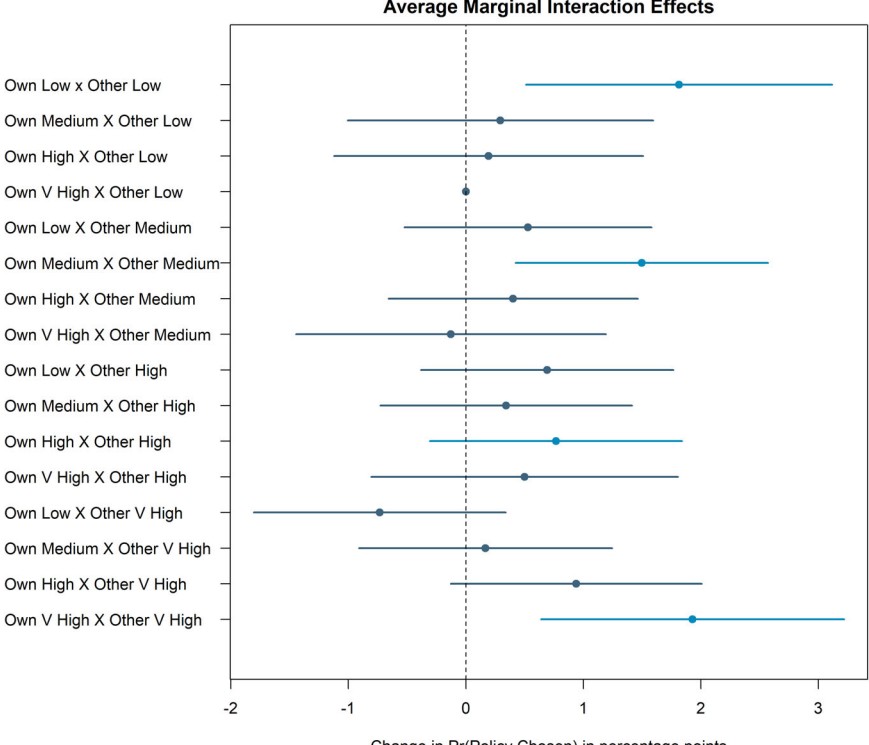

**Fig. 5 | Average marginal interaction effects of exact reciprocity (matching own and other costs) on public support in France, Germany, United Kingdom, and the United States (N = 129,280).** This plot reports average marginal interaction effects capturing the additional effect of two randomly assigned cost level features co-occurring on the probability that a policy is chosen, above and beyond their individual effects. Error bars indicate 95% confidence intervals.

is chosen of two features co-occurring, above and beyond their individual effects.

Figure 5 shows the AMIEs which capture how a specific combination of own-other country cost levels affect policy support. The baseline category is a situation where own costs are at their highest while the costs for other countries are at their lowest. We find that the interaction effects are largest when a scenario is exactly reciprocal, namely when the other countries' average costs exactly matches the respondent's domestic climate contribution. When the own and other household costs differ, their relative size has little or no impact on respondents' likelihood of selecting a policy. This lends additional support to the importance of reciprocity in the context of multilateral climate action.

## Discussion

Tremendous resources have been invested in forging international cooperation to address climate change, with limited success to this point. Across the world, it is clear that there are major distributional and domestic political obstacles to reducing greenhouse gas emissions. In such an environment, it is a fair question whether the focus on international agreements and multilateral cooperation more generally is time well spent. In this paper, we have revisited one motivation for securing such multilateral arrangements—building a robust domestic political coalition in support of costly climate action. Our finding that multilateralism is an important driver of public support for climate action has several implications. First, in terms of policymaking, this suggests that addressing climate change through international cooperation rather than exclusively unilaterally may be more important to increase public approval than previously thought. After all, effective climate action will require individuals to tolerate carbon pricing and multilateral efforts may generate the goodwill necessary to make costly climate action palatable. Second, we believe that since mass preferences over climate

action are sensitive to the policy efforts of other countries, existing theories of international interdependence that build on models of collective action can improve our understanding of public attitudes toward costly climate action. Third, the multilateral policy conjoint experiment developed here can be applied to study in more detail the relative importance of domestic and multilateral policy design features for explaining mass support for policies meant to address important sustainability challenges such as biodiversity loss, deforestation, air pollution, renewable energy transitions, waste disposal, or ocean acidification.

## Methods

### Sample

We fielded our survey in four major economies (France, Germany, the United Kingdom, and the United States). The survey was conducted online by YouGov on representative samples of the adult populations. YouGov employs matched sampling in which respondents are selected from participants in YouGov's online panel. Matched sampling involves taking a stratified random sample of the target population and then matching available internet respondents to the target sample using propensity scores. The propensity score model included age, gender, years of education, and region for the European countries and gender, age, race/ethnicity, region, and education for the United States.

United States: The field period was December 18, 2018 to January 3, 2019. The sampling frame for the target population was constructed from the full 2016 American Community Survey. All matched respondents were then assigned weights stratified on 2016 presidential vote, age, sex, race, and education to correct for remaining imbalances. The final number of observations was 4075.

France, Germany, United Kingdom: The field period was March 31, 2019 to April 04, 2019. The sampling frames for the target populations were constructed from the 2018 Eurobarometer survey with selection within strata by weighted sampling with replacements (using the

person weights on the public use file). The final number of observations was 2000 for France, 2000 for Germany, and 2000 for the United Kingdom.

Supplementary Table 1 reports the distributions of socio-demographic characteristics in the population, the raw samples, and the weighted samples by country.

**Measuring expectations about effectiveness and sustainability**
We measured expectations about the effectiveness of a carbon tax using the following question:

"In addition, if this policy is implemented by [COUNTRY, COUNTRY and other major economies], which of the following statements below do you think are true? Will this…
… provide better life for children and grandchildren
… save many plant and animal species from extinction
… improve people's health
… lead to more government regulation
… cause energy prices to rise
… cost jobs and harm the economy
… help with distributing the costs of climate change more fairly.
The value of the randomly manipulated part of the question matched the assignment in the multilateral vignette experiment described in the main text. COUNTRY was replaced with the respondent's country name.

**Reporting summary**
Further information on research design is available in the Nature Research Reporting Summary linked to this article.

## Data availability
The data used in this study are available at the Harvard Dataverse as Bechtel, Michael; Scheve, Kenneth; van Lieshout, Elisabeth, 2022, "Replication Data for: Improving Public Support for Climate Action Through Multilateralism", https://doi.org/10.7910/DVN/GCNO77, Harvard Dataverse, V1. The following software has been used to process the data: Excel (MacOS 16.65), Stata 16, and R.

## Code availability
The code used in this study is available at the Harvard Dataverse as Bechtel, Michael; Scheve, Kenneth; van Lieshout, Elisabeth, 2022, "Replication Data for: Improving Public Support for Climate Action Through Multilateralism", https://doi.org/10.7910/DVN/GCNO77, Harvard Dataverse, V1.

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

## Acknowledgements
We thank Jordan Hale, Charles Kolstad, and audiences at King's College London, Vanderbilt University, Yale University, the Global Research in International Political Economy Seminar, the 2019 IPES Conference, the Politics of Solidarity Workshop at the University of Dusiburg-Essen, and the International Institute for Applied Systems

Analysis Speaker Series for helpful comments. We thank Emma Singh for valuable research assistance. We gratefully acknowledge financial support from the Swiss Network for International Studies and the Weidenbaum Center on the Economy, Government, and Public Policy at Washington University in St. Louis. Bechtel acknowledges support from the Swiss National Science Foundation (grant #PP00P1-139035) and the German Research Foundation under Germany's Excellence Strategy (EXC 2126/1-390838866). Scheve thanks the Institute for Research in the Social Sciences for a faculty fellowship.

## Author contributions

M.M.B., K.F.S., and E.v.L. contributed equally to this work.

## Funding

## Competing interests

The authors declare no competing interests.

## Ethics

The study was approved by the Internal Review Boards at Washington University in St. Louis (#201803178) and Stanford University (eProtocol 46325).

## Additional information

**Correspondence and requests** for materials should be addressed to Michael M. Bechtel or Kenneth F. Scheve.

