## [Peer review file · Nature Communications]

REVIEWER COMMENTS

Reviewer #1 (Remarks to the Author):

This paper uses a series of cross-national survey experiments to explore the ways in which public support for climate action is conditioned by the actions of other countries. These experiments find a number of intriguing results, all of which emphasize the importance of multilateralism in shaping climate policy action.

Overall, this is a strong article which presents important empirical results. The analysis is carefully done and the experiments thoughtfully designed. The eliminated effects analysis is a reasonable way to explore mechanisms, and improves on crude mediation analysis that other work has relied on. The conjoint is appropriately analyzed, with some sophistication that nicely tracks the paper's theory (particularly the AMIE). I recommend publication subject to only additional work contextualizing the papers' results in light of current debates.

The most significant question I have revolves around the paper's framing, and how it positions itself within existing debates over global climate policymaking. The article presents its results in opposition to a recent literature that has queried the role of free-riding as a constraint in global climate politics. Yet, my read of these findings in light of that literature is more generous. That work (Bernauer, Aklin, Mildemberger, etc.), in my read, is focused on the salience of free-riding in structuring global climate policymaking, rather than making global claims that there is no reciprocity or conditionality in national climate policymaking. However, the current article suggests the later, which is perhaps too strong a characterization of those authors' arguments.) (Since I am involved in that debate, I've signed this review. Indeed, even work that has offered puzzling findings (e.g, Tingley and Tomz) about the structure of reciprocity, still show the presence of conditionality of various forms in domestic policy preferences.

I dwell on this because the findings here don't actually seem to support the dominant free-riding paradigm in thinking about multilateral institutions. Instead, the article shows really interesting evidence about other forms of reciprocity and conditionality in the incentive structures that states face when coordinating over the climate problem. There are a number of findings here that conventional wisdom models of climate action would not have predicted. For instance, this paper has a fascinating finding in its conjoint experiment showing that respondents are sensitive to cost structures in other countries (lines 185 to 193). The authors speculate on reasons for this (e.g. other-regarding preferences), and correctly use this as evidence that actions by other countries reshapes domestic political incentives to confront the climate crisis. However, classic models of free-riding would probably predict a different result: that as other countries take on more and more of the burden of confronting the climate crisis, support in the home country would go up, since this would provide more capacity for the home country

to free-ride off the other country's costly actions. I may be thinking about this in too crude terms, but it would be good for the paper to walk the reader through these observable implications more explicitly.

Reaffirming this same point, the article's claim (lines 27-30) that "the theory of the case is that voters will be more willing to contribute to the global public goods if other countries are doing so as well because multilateral policies are more likely to be effective, i.e., able to realize important social, economic, and environmental sustainability goals, and, under these agreements, resonate with reciprocal fairness norms."...this seems like a valid conclusion from this article's empirical findings, but it does not clearly specify the priors most people working in this space might have. I can't think of any existing work on climate negotiations which arrives at this position as its core theory.

In short, I think the article would benefit from a more direct effort to characterize its results in relation to classic accounts of global climate policy being shaped by free-riding and other simple models of climate negotiations as a simple collective action problem. (Notably, the word free-riding is not used once in the article!). I don't think serious scholars would argue that there is no interdependence or conditionality in global climate policies (and certainly none of the cites here make that suggestion). Instead, the value of this paper lies in shedding new light on the structure and dynamics of this interdependence. To the degree in which these forms of reciprocity are distinct from conventional game-theoretic accounts, this deserves to be surfaced. It may well be that the authors think the evidence here, on balance, supports conventional free-riding accounts of climate institutions, but this then deserves to be surfaced explicitly.

This would be particularly helpful since the article provides some of the strongest evidence I have seen isolating the specific role of fairness norms in structuring multilateral incentives to pass climate policy, which is incredibly important to thinking about the conditions under which effective and equitable global climate policies are possible. This also includes a really important finding in the conjoint experiment - the result that interaction effects are largest when the other countries' average costs matches the respondents' country.

In sum, I think this is a well-crafted article that will likely become well-cited in debates over multilateral climate policymaking. I support its publication.

-Matto Mildenberger

Reviewer #2 (Remarks to the Author):

A very interesting read on a relevant and pressing topic. The comments provided on this article are restricted to the field of my own specialisation, namely international law. I am therefore not able to assess the methodology in critical detail.

As a lawyer, a standout question for me is how 'multilateralism' is concretely conceived in this study, and how it relates to reciprocity. It could be the case that voters look at other states' policies, while the states themselves do not participate in any form of multilateralism / cooperation.

Perhaps reference to the shared global temperature goal in the Paris Agreement may be of value for the conceptual framework. More specific comments can be found attached as annotations in the article PDF.

Reviewer #3 (Remarks to the Author):

* Summary

This paper investigates the effect of multilateralism on public support for climate policy (a carbon tax). In doing so, it seeks to evaluate whether building cooperation across countries increases public support for these policies.

My review will be pretty short. I enjoyed reading the paper a lot; the design is clear and the question is interesting.

* Main comments

(1) Interpretation of the results

The empirical analysis is, as far as I can tell, done cleanly. My main concern lies in the implication and interpretation of the results. The authors contrast their study against a few recent publications and state (p.3): "these studies suggest that public support costly climate policies does not meaningfully depend on whether other countries are also contributing or not."

There are two things that I think need clarification.

First, while the four papers cited differ in what they say and do, I don't think any of these say that public support is independent on what happens elsewhere. For instance, the Baiser-McGrath and Bernauer paper find that "citizens ... have strong preferences over the design of international climate agreements and contributions of other countries to the global effort" (from their abstract). Likewise, Aklin and Mildemberger focus on free-riding and mutual defection and don't claim that there is no interdependence at all. I'm not saying that this is a straw man, but I see more connections between this set of papers and this manuscript. For instance, the authors' own results on norms (Figure 2) are not inconsistent with these earlier findings. I'm not saying that they are perfectly consistent either, but I don't quite see them as contradicting each other.

Second, let's assume that the authors disagree with my previous point and insist on contrasting multilateralism against unconditional action (fair enough!). I then don't see their results as particularly supportive of the multilateralism hypothesis. Focusing on the first set of the results, we find that 53% are unconditionally supportive and this number only goes up by a mere 6% when multilateralism is introduced. The authors argue that the 53% "is merely a slim majority that can be broken by relatively small shifts in public support." But what's the size of those who are conditional cooperators? Based on these results, it seems that it's a small groups (certainly smaller than the 53% by quite a margin).

This is a long and convoluted way to say: there is a difference between saying that multilateralism helps increase support (which the authors show convincingly) and that it reflects the way a large segment of the population thinks (where it seems to me that the authors show the opposite).

(2) Conjoint

I have two questions on the conjoint analysis.

First, I would like to hear more from the authors about the results regarding the cost level for other countries (Figure 3). If I read this correctly, respondents became less supportive the more others had to pay. I'm trying to think about how to square this with the idea what we generally prefer if others pay more. Is this finding telling us about how people evaluate these costs? Or, alternatively, did readers misunderstand the prompt? (More generally, it wasn't clear to me whether the authors included comprehension checks or not.)

Second, in appendix (p.1), the authors say that they interviewed 2,000 respondents per country (except in the US, where they interviewed 4,081 people). They also say that they repeated the conjoint 8 times

for each respondent. But Figure A6 says that the sample size is 32,000 for France, Germany, and the UK. Shouldn't it be half (2,000*8 tasks)? Am I missing something?

* Minor comments

p.5 "To explore the role of effectiveness in the eliminated effects framework..."

- I found the wording of that paragraph a bit opaque and hard to follow (even though I'm familiar with the Acharya et al piece!). Maybe it's worth editing it for readers who are not familiar with this approach (or expand in the methods section)? Maybe a figure to clarify what is being compared to what? I don't have a good suggestion, but given that these are important results, it would be good to have a clear explanation.

Improving Public Support for Climate Action Through Multilateralism *

Michael M. Bechtel
Washington University in St. Louis
mbechtel@wustl.edu

Kenneth Scheve
Yale University
kenneth.scheve@yale.edu

Elisabeth van Lieshout
Stanford University
lieshout@stanford.edu

September 2021

*We thank Jordan Hale, Charles Kolstad, and audiences at Yale University, the Global Research in International Political Economy Seminar, and the 2019 IPES Conference for helpful comments. We thank Emma Singh for valuable research assistance. We gratefully acknowledge financial support from the Swiss Network for International Studies and the Weidenbaum Center on the Economy, Government, and Public Policy at Washington University in St. Louis. Bechtel acknowledges support from the Swiss National Science Foundation (grant #PP00P1-139035) and Scheve thanks the Institute for Research in the Social Sciences for a faculty fellowship.

Abstract

For decades, policymakers have been attempting to negotiate multilateral climate agreements. One of the motivations for securing cooperation among multiple states is the belief that the public will be more supportive of adopting costly climate policies if other countries do so, both because this makes it more likely that important sustainability goals will be reached and because those efforts resonate with widely held fairness norms. However, some recent research suggests that public approval of climate action is independent of the policy choices made by other countries. Here, we present two different experimental studies fielded in multiple countries showing that multilateralism significantly increases public approval of costly climate action. Multilateralism makes climate policy more appealing by improving sustainability beliefs and the policy's perceived fairness. Pursuing climate action within a multilateral setting does not only promise improved policy impacts, but may also generate higher levels of public support.

Millions of people around the world have mobilized to protest the failure of governments
to implement policies that would substantially reduce greenhouse gas emissions and ad-
dress the climate challenge.¹ It would seem that these concerns resonate not just with
activists, but the majority of voters in many of the major emitter countries.² Nonetheless,
often when governments have acted to reduce emissions, they have been met by intense
opposition, some of it certainly by organized special interest groups but some of it by
voters reluctant to incur the costs of higher carbon prices.³

How can we explain political opposition to costly climate action? One part of the an-
swer is surely the scale of the distributional conflict associated with the sort of energy
transition necessary to address the climate challenge. The costs and benefits of climate
change and climate policy vary across individuals, generations, firms, industries, regions,
and countries (Bechtel, Genovese and Scheve, 2017; Bayer and Genovese, 2020; Kennard,
2020; Cory, Lerner and Osgood, 2021; Colgan, Green and Hale, 2021). Even in surveys
that initially seem to indicate that large majorities favor climate policy action, making the
costs and their distribution explicit can quickly erode support. Another common expla-
nation points to the global public goods character of addressing climate change (Barrett,
2003; Stavins, 2011; Keohane, 2015; Underdal, 2017; Dolšak and Prakash, 2018; Nordhaus,
2019). Sustainable greenhouse gas emissions are a global public good with the usual char-
acteristics that may lead to underprovision. Policymakers and researchers have invested
significantly in identifying international agreements and institutions that would address

¹Examples include large-scale social movements such as Fridays for Future. See *Protesting Climate Change, Young People Take to Streets in a Global Strike*, The New York Times, <https://www.nytimes.com/2019/09/20/climate/global-climate-strike.html>, last accessed on Nov 27th, 2019; *Global warning: climate protests around the world—in pictures*, The Guardian, <https://www.theguardian.com/environment/gallery/2019/oct/09/international-rebellions-to-save-the-planet-in-pictures>, last accessed on Nov 27th, 2019.

²“Climate crisis seen as ‘most important issue’ by public, poll shows”, *The Guardian*, <https://www.theguardian.com/environment/2019/sep/18/climate-crisis-seen-as-most-important-issue-by-public-poll-shows>; Gallup, “Environment”, <https://news.gallup.com/poll/1615/environment.aspx>.

³“Hundreds of Thousands in France Protest Taxes by Blocking Roads”, *New York Times*, <https://www.nytimes.com/2018/11/17/world/europe/french-drivers-protest-fuel-taxes.html?searchResultPosition=17>.

the international component of the problem. This practical and scholarly effort builds on
insights into how small and large communities across many diverse issue domains solve
public goods problems (Ostrom, 1990; Taylor, 1987; Axelrod, 1984). One of the many mo-
tivations behind policymakers' interests in creating a workable international framework
for cooperation on climate change is to secure greater public support for policies to reduce
greenhouse gas emissions, especially policies that are costly to their voter base. The the-
ory of the case is that voters will be more willing to contribute to the global public good
if other countries are doing so as well because multilateral policies are more likely to be
effective, i.e., able to realize important social, economic, and environmental sustainability
goals, and, under these agreements, resonate with reciprocal fairness norms.

In this paper, we investigate whether multilateralism increases support for costly cli-
mate policies. Multilateralism may be thought of as describing a continuum of behav-
ior in international politics. This continuum may range from decentralized cooperation
among states, e.g., a set of two or more countries independently contributing to a public
good in an ad-hoc fashion (Cranmer, Heinrich and A.Desmarais, 2014), to highly insti-
tutionalized forms of interaction in and through international organizations (Milner and
Tingley, 2013; Hawkins et al., 2006). We study multilateralism in the context of climate co-
operation where countries engage in "decentralized policy coordination" (Keohane and
Victor, 2016, p. 570) which typically entails declaratory statements and agreements that
specify policy objectives and possible means, but lack enforcement mechanisms. Early
public opinion on climate agreements found that publics value broad international par-
ticipation and multilateral climate action (Tvinnereim and Lachapelle, 2016; Tingley and
Tomz, 2014; Bechtel and Scheve, 2013), but more recent work suggests that explanations
of public opinion drawing on the public goods framework and multilateral policy ap-
proaches may not be the most useful accounts of the mass politics of climate change
(Aklin and Mildenerger, 2020; Mildenerger, 2019; Beiser-McGrath and Bernauer, 2019;
Gampfer, Bernauer and Kachi, 2014). Instead, these studies suggest that public support

for costly climate policies does not meaningfully depend on whether other countries are
also contributing or not. We present evidence from two new experimental studies in four
countries—France, Germany, the United Kingdom, and the United States—that suggests
that voters care about the policies of other countries and are more likely to support costly
policies when other countries are also doing so. We present evidence that this interdepen-
dence is due to how multilateralism shapes the effectiveness of policies and its resonance
with reciprocal fairness norms.

**The Causal Effect of Multilateralism on Climate Policy Sup-** 56 **port**

To assess whether multilateralism causes higher levels of policy support, we start by em-
ploying a vignette survey experiment in France, Germany, and the United Kingdom. We
conduct the experiment as a module in original surveys fielded in April 2019. For each
country, the samples are representative of the adult population with 2,000 respondents
in each. Appendix A provides a detailed description of the sampling frame.⁴ Appendix
Table A.1 offers a comparison of the distribution of sociodemographic characteristics in
the target population, the raw sample, and the weighted sample. All our results employ
survey weights, but the findings are very similar when analyzing the unweighted data.

To investigate the effect of multilateralism on support for climate policy, we focus on
opinion regarding a hypothetical carbon tax. This policy instrument is relatively easy for
respondents to understand and it is reasonable to expect it to reduce carbon emissions.
Critically, it can be implemented by either a single country or multiple countries. The
exact wording of the vignette experiment is:

⁴The survey instrument is part of the replication archive for this study which will be made available at the Harvard Dataverse before publication. There are no restrictions on data availability. The study was approved by the Internal Review Boards at Washington University in St. Louis (#201803178) and Stanford University (eProtocol 46325). Informed consent was obtained from the survey participants.

“Suppose COUNTRY [decides, and other major economies decide] to implement a car-
bon tax, which is an additional tax on the CO2 content of fuels, to address climate change.
Generally speaking, do you approve or disapprove of COUNTRY implementing such
policies?”

We randomize whether a respondent receives a version of this question in which the car-
bon tax would be implemented unilaterally or whether it would be part of a multilateral
setting in which other major economies also introduce a carbon tax. We record responses
on a 1-10 (strongly approve-strongly disapprove) answer scale.

Higher levels of international participation in climate action may increase public sup-
port because of heightened expectations about the effectiveness of these policy efforts.
We investigate the effectiveness mechanism by crossing the multilateralism experiment
with a second vignette experiment that provided information about expected policy im-
pacts. This allows us to analyze the causal mechanism in an eliminated-effects-framework
(Acharya, Blackwell and Sen, 2018). Taken together, the experiment consisted of one con-
trol group – which received no additional information – and two treatment groups (*Effec-*
*tiveness: Low* and *Effectiveness: High*):

*Effectiveness: Low*: “Most experts think this will avoid a few of the economically and en-
vironmentally damaging consequences of climate change.”

*Effectiveness: High*: “Most experts think this will avoid most of the economically and en-
vironmentally damaging consequences of climate change.”

We analyze the proportion of individuals approving the introduction of a carbon tax
(levels of support that exceed 5, the midpoint of the scale). Figure 1a reports carbon
tax support by randomly assigned multilateralism condition along with 95% confidence
intervals. In the control condition that did not provide any effectiveness information, we
find that about 53% of all respondents support the introduction of a carbon tax if this
policy is pursued unilaterally. It is noteworthy, that this is merely a slim majority that can
be broken by relatively small shifts in public support. However, when other countries also
decide to introduce a carbon tax, support is 59%, which is equivalent to a 6 percentage
points increase over the unilateralism condition.

To explore the role of effectiveness in the eliminated effects framework, first recall that
the estimate of the effect of the multilateralism condition for the control group that did
not receive information about effectiveness is our estimate of the average treatment effect.
The estimate of the effect of multilateralism for respondents exposed to the low or high
effectiveness condition is our estimate of the average controlled direct effect which is that
part of the total effect that is not due to mediation by or interaction with effectiveness. The
average treatment effect minus the average controlled direct effect is the eliminated effect
(Acharya, Blackwell and Sen, 2018). This quantity is the portion of the average treatment
effect that can be explained by the effect of the treatment–multilateralism–through the
mediator–effectiveness–and any interaction between multilateralism and effectiveness.

Figure 1a shows the results for the low and high effectiveness conditions in which we
fix respondents’ beliefs about policy effectiveness. We find that in the high effectiveness
condition, a unilateral carbon tax approach is backed by about 57% and that proportion
increases to 62% in the multilateralism treatment. In contrast, when policy effectiveness
is low, the switch from a unilateral to a multilateral climate policy framework has almost
no effect on carbon tax approval.

Figure 1b shows that multilateralism causes a significant increase in carbon tax sup-
port in the control and in the high effectiveness condition, but not in the low effectiveness
condition. This means our two eliminated effect estimates provide contrasting evidence
on how well effectiveness explains the impact of multilateralism on public support. For
high effectiveness, the eliminated effect is very close to zero while for low effectiveness, it
is positive at about 6 percentage points and close to being significant at the 5%-level. The
latter estimate indicates that most of the effect of multilateralism is explained by effec-
tiveness, but of course the caveat is that high effectiveness estimate is inconsistent with
this interpretation.

Figure 1: The Effect of Multilateralism on Carbon Tax Support (N=6,000)

(a) Proportion Supporting a Carbon Tax by Multilateralism and Effectiveness Prime

(b) Causal and Eliminated Effects (EE) of Multilateralism by Effectiveness Prime

Note: (a) The figure shows the proportion of individuals supporting the introduction of a carbon tax by randomly assigned multilateralism and effectiveness conditions. (b) The figure reports the average treatment effect (ATE), average controlled direct effects (ACDE), and eliminated effects (EE) of multilateralism by randomly assigned effectiveness condition (High=High Effectiveness, Low=Low Effectiveness) estimated using a linear probability model with robust standard errors. Regressions control for gender, age, income, education, and employment status. Country fixed effects included. Error bars indicate 95% confidence intervals. N(France)=2,000, N(Germany)=2,000, N(United Kingdom)=2,000. Unweighted results are very similar, see Figure A.1.

The Multilateralism Effect and Beliefs about Realizing Sustainability Goals

To further explore which impacts individuals expect from a multilateral as opposed to a unilateral climate policy framework, we added a question after the multilateralism vignette experiment that prompted respondents to indicate whether they thought specific statements about the environmental, economic, and fairness benefits and costs of climate action were true or false.⁵

We estimate how multilateralism affects sustainability beliefs by regressing whether a statement is selected as true on a multilateralism treatment indicator, a full set of sociodemographic control variables, and country fixed effects. The results in Figure 2 indicate that multilateralism matters: individuals are significantly more likely to expect greater environmental benefits, lower economic and governance costs, and fairer cost distributions from multilateral efforts. Regarding benefits, we find that multilateralism significantly improves beliefs about the ability to reach important sustainability goals, for example, the potential for climate action to improve the lives of respondents' children and grandchildren. We find broadly similar effects when analyzing whether the policy will save endangered animals and plants and whether it will improve public health outcomes.

When assessing the impact of multilateralism on costs, we find that multilateralism does not systematically affect concerns related to increased regulation and potential job losses caused by progressive climate action. However, multilateral policy reduces concerns about energy price increases. Finally, multilateralism has a significant and strong impact on respondents' beliefs about whether the costs of climate action will be dis-

⁵The question wording was "In addition, if this policy is implemented by [COUNTRY, COUNTRY and other major economies], which of the following statements below do you think are true? Will this ... provide better life for children and grand children ... save many plant and animal species from extinction ... improve people's health ... lead to more government regulation ... cause energy prices to rise ... cost jobs and harm the economy ... help with distributing the costs of climate change more fairly." The value of the randomly manipulated part of the question matched the assignment in the vignette experiment described above.

Figure 2: The Causal Effects of Multilateralism on Sustainability Beliefs: Benefits, Costs, and Fairness of Climate Action in France, Germany, and the United Kingdom (N=6,000)

Note: This plot reports coefficients from linear regressions of statement approval on a binary indicator that is one if climate action is multilateral and is zero if climate action is unilateral. Error bars indicate 95% confidence intervals. All regressions control for gender, age, income, education, and employment status. Country fixed effects included. Survey weights applied. Results for the unweighted data are very similar, see Figure A.3. N(France)=2,000, N(Germany)=2,000, N(United Kingdom)=2,000.

tributed more fairly.

Multilateral Policy Design and Climate Support

Our second study examines the effect of multilateralism on support for climate policies
 by investigating whether individuals have preferences  the policies in other coun-
 tries and whether those policies influence the willingness of respondents to adopt costly
 policies in their own country. This study was conducted as a separate module in the
 French, German, and UK surveys described above and in an additional survey fielded in

the United States in December 2018 (see Appendix Section A for further description).

To study these questions, we constructed a randomized conjoint experiment that pre-

[revised manuscript text omitted]

be called exact reciprocity in which individuals are more supportive of adopting climate
policies that match the efforts of other countries. Exact matching is a natural focal point
for reciprocal behavior.

We test this expectation by investigating whether scenarios in which other countries
match the cost levels of one's own country increases support for climate action more
strongly than non-matching contributions. We compute the average marginal interaction
effects (AMIEs) as defined by Egami and Imai (2019). The AMIE quantity of interest

⁶The interdependence between domestic and foreign carbon pricing is all the more noteworthy since it has proven challenging to detect theoretically meaningful interactions between attribute features in conjoint data (Ratkovic and Tingley, 2017).

Figure 4: The Causal Effects of Climate Policy Costs by Other Costs in France, Germany, United Kingdom, and the United States (N=129,280)

(b) Difference in Effect of Own Cost if Other Costs Increase from Very Low to Very High

Note: This plot reports coefficients from linear regressions of policy approval on randomly assigned climate policy household costs introduced in other countries. Error bars indicate 95% confidence intervals. Results for the unweighted data are very similar (see Figure A.8). N(France)=32,000, N(Germany)=32,000, N(United Kingdom)=32,000, N(United States)=33,280.

captures the additional effect on the probability that a policy is chosen of two features
 co-occurring, above and beyond their individual effects.

Figure 5 reports the estimated AMIEs. The baseline category is a situation where own
 costs are at their highest while the costs for other countries are at their lowest. We find
 that the interaction effects are largest when a scenario is exactly reciprocal, namely when
 the other countries' average costs exactly matches the respondent's domestic climate con-
 tribution. When the own and other household costs differ, their relative size has little or
 no impact on respondents' likelihood of selecting a policy. This lends additional support
 to the importance of reciprocity in the context of multilateral climate action.

Figure 5: Average Marginal Interaction Effects of Own and Other Costs on Public Support in France, Germany, United Kingdom, and the United States(N=129,280)

Discussion

Tremendous resources have been invested in forging international cooperation to address
climate change with limited success to this point. Across the world, it is clear that there
are major distributional and domestic political obstacles to reducing greenhouse gas emis-
sions. In such an environment, it is a fair question to wonder whether the focus on inter-
national agreements and cooperation more generally is time well spent. In this paper, we
have revisited one motivation for securing such an agreement—building a robust political
coalition in support of costly climate policies.

Our finding that multilateralism is an important driver of public support for climate
action have several implications. First, in terms of policymaking, this suggest that ad-
dressing climate change through international cooperation rather than exclusively uni-
laterally may be more important to increase public approval than previously thought.
After all, effective climate action will require individuals to tolerate carbon pricing and
multilateral efforts may generate the goodwill necessary to make costly climate action
palatable. Second, given that mass preferences over climate action mirror international
interdependence as well as factors related to domestic distributive conflict, our results
suggest that theories of international interdependence that build on models of collective
action can improve our understanding of public attitudes toward public policy. Third,
the multilateral policy conjoint experiment developed here can be applied to study in
more detail the relative importance of domestic and multilateral policy features for ex-
plaining mass support for policies meant to address important sustainability challenges
such as biodiversity loss, deforestation, air pollution, renewable energy transitions, waste
disposal, or ocean acidification.

**Data Availability Statement**

Data and replication materials will be made available at the Harvard Dataverse before
publication.

**Code Availability Statement**

Statistical code will be made available as part of the replication materials at the Harvard
Dataverse before publication.

**References**

- Acharya, Avidit, Matthew Blackwell and Maya Sen. 2018. "Analyzing Causal Mechanism in Survey Exper-
iments." *Political Analysis* 26:357–378.
- Aklin, Michael and Matto Mildenerger. 2020. "Prisoners of the Wrong Dilemma: Why Distributive Con-
flict, Not Collective Action, Characterizes the Politics of Climate Change." *Global Environmental Politics*
20(4):4–27. SSRN Working Paper, <http://dx.doi.org/10.2139/ssrn.3281045>.
- Aldy, Joseph E., Alan J. Krupnick, Richard G. Newell, Ian W. H. Parry and William A. Pizer. 2010. "De-
signing Climate Mitigation Policy." *Journal of Economic Literature* 48(4):903–934.
- Axelrod, Robert. 1984. *The Evolution of Cooperation*. New York: Basic Books.
- Barrett, Scott. 2003. *Environment and Statecraft: The Strategy of Environmental Treaty-Making*. Oxford: Oxford
University Press.
- Bayer, Patrick and Federica Genovese. 2020. "Beliefs About Climate Action Consequences Under Weak
Global Institutions: Sectors, Home Bias, and International Embeddedness." *Global Environmental Politics*
20(4):28–50.
- Bechtel, Michael M., Elisabeth van Lieshout and Kenneth F. Scheve. 2020. "Constant carbon pricing in-
creases support for climate action compared to ramping up costs over time." *Nature Climate Change*
p. September.

- Bechtel, Michael M., Federica Genovese and Kenneth F. Scheve. 2017. "Interests, Norms and Support for the
Provision of Global Public Goods: The Case of Climate Co-operation." *British Journal of Political Science*
pp. 1–23.
- Bechtel, Michael M. and Kenneth F. Scheve. 2013. "Mass Support for Global Climate Agreements Depends
on Institutional Design." *Proceedings of the National Academy of Sciences* 110(32):13763–13768.
- Beiser-McGrath, Liam F. and Thomas Bernauer. 2019. "Commitment Failures Are Unlikely to Undermine
Public Support for the Paris agreement." *Nature Climate Change* 9(3):248–252.
- Colgan, Jeff D., Jessica F. Green and Thomas N. Hale. 2021. "Asset Revaluation and the Existential Politics
of Climate Change." *International Organization* 75(2):586–610.
- Cory, Jared, Michael Lerner and Iain Osgood. 2021. "Supply Chain Linkages and the Extended Carbon
Coalition." *American Journal of Political Science* 65(1):69–87.
**URL:** <https://onlinelibrary.wiley.com/doi/abs/10.1111/ajps.12525>
- Cranmer, Skyler J., Tobias Heinrich and Bruce A. Desmarais. 2014. "Reciprocity and the structural determi-
nants of the international sanctions network." *Social Networks* (5-22).
- Dolšak, Nives and Aseem Prakash. 2018. "The Politics of Climate Change Adaptation." *Annual Review of*
*Political Science* 43:317–341.
- Egami, Naoki and Kosuke Imai. 2019. "Causal Interaction in Factorial Experiments: Application to Conjoint
Analysis." *Journal of the American Statistical Association* 114(526):529–540.
**URL:** <https://doi.org/10.1080/01621459.2018.1476246>
- Gampfer, Robert, Thomas Bernauer and Aya Kachi. 2014. "Obtaining Public Support for North-South Cli-
mate Funding: Evidence from Conjoint Experiments in Donor Countries." *Global Environmental Change*
29:118–126.
- Hawkins, Darren G., David A. Lake, Daniel L. Nielson and Michael J. Tierney. 2006. *Delegation and Agency*
*in International Organizations*. Cambridge (Mass.): Cambridge University Press.
- Kennard, Amanda. 2020. "The Enemy of My Enemy: When Firms Support Climate Change Regulation."
*International Organization* 74(2):187–221.
- Keohane, Robert O. 2015. "The Global Politics of Climate Change: Challenge for Political Science." *PS:*
*Political Science and Politics* 48(1):19–26.

- Keohane, Robert O. and David Victor. 2016. "Cooperation and Discord in Global Climate Policy." *Nature*
*Climate Change* 6:570–575.
- Mildenerger, Matteo. 2019. "Support for Climate Unilateralism." *Nature Climate Change* 9:187–190.
- Milner, Helen V. and Dustin Tingley. 2013. "The Choice for Multilateralism: Foreign Aid and American
Foreign Policy." *Review of International Organizations* 8(3):313–341.
- Nordhaus, William. 2019. "Climate Change: The Ultimate Challenge for Economics." *American Economic*
*Review* 109(6):1991–2014.
- Ostrom, Elinor. 1990. *Governing the Commons: The Evolution of Institutions for Collective Action*. Cambridge
University Press.
- Ratkovic, Marc and Dustin Tingley. 2017. "Sparse Estimation and Uncertainty with Application to Sub-
group Analysis." *Political Analysis* 25(1):1–40.
- Stavins, Robert N. 2011. "The Problem of the Commons: Still Unsettled after 100 Years." *American Economic*
*Review* 101(1):81–108.
- Taylor, Michael. 1987. *The Possibility of Cooperation*. Cambridge: Cambridge University Press.
- Tingley, Dustin and Michael Tomz. 2014. "Conditional Cooperation and Climate Change." *Comparative*
*Political Studies* 47(3):344–368.
- Tvinnereim, Endre and Erick Lachapelle. 2016. "Is Support for International Climate Action Conditional
on Perceptions of Reciprocity? Evidence from Three Population-based Survey Experiments in Canada,
the US, and Norway." *Cosmos* 12(1):43–55.
- Underdal, Arild. 2017. "Climate Change and International Relations (After Kyoto)." *Annual Review of Polit-*
*ical Science* 20:169–188.

Online Appendix for
“Improving Public Support for Climate Action Through
Multilateralism”

**A Description of Climate Policy Survey (France, Germany,
United Kingdom, United States, N=10,081)**

We fielded our survey in four major economies (France, Germany, the United Kingdom, and the United
States). The survey was conducted online by YouGov on representative samples of the adult populations.
YouGov employs matched sampling in which interviews are conducted from participants in YouGov’s on-
line panel. Matched sampling involves taking a stratified random sample of the target population and then
matching available internet respondents to the target sample using propensity scores. The propensity score
model included age, gender, years of education, and region for the European countries and gender, age,
race/ethnicity, region, and education for the United States. The study was approved by the Internal Re-
view Boards at Washington University in St. Louis (#201803178) and Stanford University (eProtocol 46325).

*United States:* The field period was December 18, 2018 to January 3, 2019. The sampling frame for the target
population was constructed from the full 2016 American Community Survey. All matched respondents
were then assigned weights stratified on 2016 presidential vote, age, sex, race, and education to correct for
remaining imbalances. The final number of observations was 4,081.

*France, Germany, United Kingdom:* The field period was March 31, 2019 to April 04, 2019. The sampling
frames for the target populations were constructed from the 2018 Eurobarometer survey with selection
within strata by weighted sampling with replacements (using the person weights on the public use file).
The final number of observations was 2,000 for France, 2,000 for Germany, and 2,000 for the United King-
dom.

Table A.1 reports the distributions of sociodemographic characteristics in the population, the raw samples,
and the weighted samples by country.

**B Appendix Tables**

Table A.1: Climate Policy Survey: Distribution of Socio-Demographics in the Target Population, the Raw Sample, and the Weighted Sample by Country (Total N=10,081)

United States			
	Population	Raw Sample	Weighted Sample
Age: 18-34	30	27	30
Age: 35-49	25	23	25
Age: 50-64	25	30	25
Age: 65+	20	22	20
Education: Less than High School	12	7	12
Education: High School Degree	28	29	28
Education: Associate's Degree or Some College	31	32	31
Education: BA or higher	29	32	29
Gender: Male	48	47	49
Gender: Female	51	53	51
Germany			
	Population	Raw Sample	Weighted Sample
Age: 18-29	19	18	19
Age: 30-44	21	21	21
Age: 45-64	35	24	35
Age: 65+	24	25	25
Education: 16yrs or less	38	43	38
Education: 17-18	19	32	19
Education: 19+	43	25	43
Gender: Male	49	48	48
Gender: Female	51	51	51
France			
	Population	Raw Sample	Weighted Sample
Age: 18-29	20	17	20
Age: 30-44	23	25	23
Age: 45-64	32	36	32
Age: 65+	26	22	26
Education: 16yrs or less	26	12	26
Education: 17-18	25	48	26
Education: 19+	49	40	48
Gender: Male	47	46	47
Gender: Female	53	54	53
United Kingdom			
	Population	Raw Sample	Weighted Sample
Age: 18-29	22	19	22
Age: 30-44	26	27	26
Age: 45-64	30	32	30
Age: 65+	22	23	22
Education: 16yrs or less	41	32	41
Education: 17-18	28	21	20
Education: 19+	31	47	38
Gender: Male	50	46	50
Gender: Female	50	55	50

C Appendix Figures

Figure A.1: The Effect of Multilateralism on Carbon Tax Support (N=6,000), Unweighted

(a) Proportion Supporting a Carbon Tax by Multilateralism and Effectiveness Prime

(b) Causal and Eliminated Effects (EE) of Multilateralism by Effectiveness Prime

Note: (a) The figure shows the proportion of individuals supporting the introduction of a carbon tax by randomly assigned multilateralism and effectiveness conditions. (b) The figure reports the causal effects of multilateralism by randomly assigned effectiveness condition (control=no information, Eff: High=High Effectiveness, Eff: Low=Low Effectiveness) along with the eliminated effects (EE) estimated using a linear probability model with robust standard errors. Regressions control for gender, age, income, education, and employment status. Country fixed effects included. Error bars indicate 95% confidence intervals. N(France)=2,000, N(Germany)=2,000, N(United Kingdom)=2,000.

Figure A.2: The Effect of Multilateralism on Carbon Tax Support (N=6,000), by Country

(a) Proportion Supporting a Carbon Tax by Multilateralism and Effectiveness Prime

(b) Causal Effects of Multilateralism by Effectiveness Prime

Note: (a) The figure shows the proportion of individuals supporting the introduction of a carbon tax by randomly assigned multilateralism and effectiveness conditions. (b) The figure reports the causal effects of multilateralism by randomly assigned effectiveness condition (control=no information, Eff: High=High Effectiveness, Eff: Low=Low Effectiveness) estimated using a linear probability model with robust standard errors. Error bars indicate 95% confidence intervals. Survey weights applied. N(France)=2,000, N(Germany)=2,000, N(United Kingdom)=2,000.

Figure A.3: The Causal Effects of Multilateralism on Expectations about the Benefits, Costs, and Fairness of Climate Action in France, Germany, and the United Kingdom (N=6,000), Unweighted

Note: This plot reports coefficients from linear regressions of statement approval on a binary indicator that is one if climate action is multilateral and is zero if climate action is unilateral. Error bars indicate 95% confidence intervals. Without use of survey weights. N(France)=2,000, N(Germany)=2,000, N(United Kingdom)=2,000.

Figure A.4: The Causal Effects of Multilateralism on Expectations about the Benefits, Costs, and Fairness of Climate Action in France, Germany, and the United Kingdom (N=6,000), by Country

Note: This plot reports coefficients from linear regressions of statement approval on a binary indicator that is one if climate action is multilateral and is zero if climate action is unilateral. Error bars indicate 95% confidence intervals. Survey weights applied. N(France)=2,000, N(Germany)=2,000, N(United Kingdom)=2,000.

Figure A.5: Instructions and Example Profile for the Climate Multilateralism Conjoint Experiment

We will now provide you with several scenarios which describe a set of policies for the United States that will impact climate change and information about what other major economies are doing. The scenarios will vary in how costly they are to households, how those costs will change over time, whether investments will be made in mitigation efforts to reduce greenhouse gas emissions thus making global warming less likely or in adaptation efforts to adjust to environmental change to lessen the negative effects of global warming. These dimensions will vary both for the United States and in other major economies.

For each comparison we would like you to tell us which of the scenarios you prefer. You may like several alternatives similarly or may not like either of them at all. Regardless of your overall evaluation, please indicate which alternative you prefer.

In total, we will show you 8 comparisons. People have different opinions about this issue and there are no right or wrong answers. Please take your time when reading the potential scenarios.

	Scenario 1	Scenario 2
In the United States		
Average household costs per month	\$107	\$213
Distribution of costs over time	gradually decreasing	constant over time
Adaptation (new technology, building dams, ...)	88%	2%
Mitigation (reducing greenhouse gas emissions, removing greenhouse gases from the atmosphere)	12%	98%
In other major economies		
Average household costs per month	\$53	\$107
Distribution of costs over time	gradually decreasing	gradually decreasing
Adaptation (new technology, building dams, ...)	0%	62%
Mitigation (reducing greenhouse gas emissions, removing greenhouse gases from the atmosphere)	100%	38%
Which of these scenarios do you prefer?	<input type="radio"/>	<input type="radio"/>

Note: This figure shows the instructions to respondents for the conjoint experiment and an example profile pair presented to respondents in the United States. Respondents assessed eight paired profiles. The experiment randomly varied whether the attributes for a respondent's own country or those for other major economies were listed first. For each respondent the order remained unchanged for all conjoint tasks. All attribute levels were fully and separately randomized.

Figure A.6: The Causal Effects of Climate Policy Choices by Other Countries on Public Support in France, Germany, United Kingdom, and the United States (N=129,280), Un-weighted

Note: This plot reports coefficients from linear regressions of policy approval on randomly assigned climate policy choices made by other countries. Error bars indicate 95% confidence intervals. Without use of survey weights. N(France)=32,000, N(Germany)=32,000, N(United Kingdom)=32,000, N(United States)=33,280.

Figure A.7: The Causal Effects of Climate Policy Choices by Other Countries on Public Support in France, Germany, United Kingdom, and the United States (N=129,280), by country

Note: This plot reports coefficients from linear regressions of policy approval on randomly assigned climate policy choices made by other countries. Error bars indicate 95% confidence intervals. Survey weights applied. N(France)=32,000, N(Germany)=32,000, N(United Kingdom)=32,000, N(United States)=33,280.

Figure A.8: The Causal Effects of Climate Policy Costs by Other Costs in France, Germany, United Kingdom, and the United States (N=129,280), Unweighted

(a) Effects of Own Cost, Conditional on Other Costs

(b) Difference in Effect of Own Cost if Other Costs Increase from Very Low to Very High

Note: This plot reports coefficients from linear regressions of policy approval on randomly assigned climate policy household costs introduced in other countries. Error bars indicate 95% confidence intervals. Without use of survey weights. N(France)=32,000, N(Germany)=32,000, N(United Kingdom)=32,000, N(United States)=33,280.

Figure A.9: The Causal Effects of Climate Policy Costs by Other Costs in France, Germany, United Kingdom, and the United States (N=129,280), by country

(a) Effects of Own Cost, Conditional on Other Costs

(b) Difference in Effect of Own Cost if Other Costs Very High and Own Costs if Other Costs Very Low

Note: This plot reports coefficients from linear regressions of policy approval on randomly assigned climate policy household costs introduced in other countries. Error bars indicate 95% confidence intervals. Without use of survey weights. N(France)=32,000, N(Germany)=32,000, N(United Kingdom)=32,000, N(United States)=33,280.

Figure A.10: Average Marginal Interaction Effects with Screening and Collapsing of Variables

Response to Reviewer 1

1. *“The most significant question I have revolves around the paper's framing, and how it positions itself within existing debates over global climate policymaking. The article presents its results in opposition to a recent literature that has queried the role of free-riding as a constraint in global climate politics. Yet, my read of these findings in light of that literature is more generous. That work (Bernauer, Aklin, Mildemberger, etc.), in my read, is focused on the salience of free-riding in structuring global climate policymaking, rather than making global claims that there is no reciprocity or conditionality in national climate policymaking. However, the current article suggests the later, which is perhaps too strong a characterization of those authors' arguments.) (Since I am involved in that debate, I've signed this review. Indeed, even work that has offered puzzling findings (e.g. Tingley and Tomz) about the structure of reciprocity, still show the presence of conditionality of various forms in domestic policy preferences. I dwell on this because the findings here don't actually seem to support the dominant free-riding paradigm in thinking about multilateral institutions. Instead, the article shows really interesting evidence about other forms of reciprocity and conditionality in the incentive structures that states face when coordinating over the climate problem.”*

We agree that our paper, while related to controversies about the importance of free-riding in climate politics, is designed to learn about the cross-national interdependence of public support for climate action. In other words, we are studying the effects of multilateralism on policy opinions rather than testing the global public goods model of climate politics. Admittedly, there is some divergence in our perceptions of whether there is agreement in the literature about the effects of multilateralism. We have carefully re-read our paper and made edits clarifying that we focus on the role of multilateralism for our understanding of the mass politics of climate action and that our goal is not to evaluate the freeriding prediction in the context of climate action. We further provide more information about the extent of disagreement about the role of multilateralism in the literature, although we think our study is an empirical contribution even for those scholars with a prior that multilateralism is important in shaping public opinion.

Our summary of the literature on multilateralism aims to do justice to how the authors have described their own findings. To this end, we have carefully re-read the literature on climate multilateralism. Some studies offer evidence that seems explicitly or implicitly consistent with the public valuing international cooperation on climate change (e.g., Tvinneraum/Lachapelle 2016; Tingley/Tomz 2014; Bechtel/Scheve 2013). We agree that this includes Aklin and Mildenerger (2020) who explicitly acknowledge the importance of various types of international conditionality.

However, several more recent studies make explicit claims about the limited relevance or irrelevance of the behavior of other countries for our understanding of climate policy support. For example, a study recently published in *Nature Climate Change* (Beiser-McGrath/Bernauer 2019) states that “failure of other countries to act does not, per se, reduce the public’s appetite for taking action against climate change. This is most evident in China, where the behaviour of other countries has effectively no impact on preferences regarding climate policy design. Whereas we find some effects of information on other countries’ behaviour on preferences regarding particular attributes of climate policy design, this does not lead to a substantive decrease in overall support for the most popular international climate policy.” (p. 251). This is also how others have interpreted these policy-relevant results: “Writing in *Nature Climate Change*, Liam Beiser-McGrath and Thomas Bernauer show that publics in China and the United States maintain their support for unilateral climate reforms, even when they learn that other countries are not acting” (Mildenerger 2020, p. 187).

To be fair, the key findings reported in Beiser-McGrath and Bernauer (2019 NCC) are based on an erroneous interpretation of their forced-choice conjoint data. This is because in this data, the baseline level of support for an agreement is always 50% as an artifact of their design. Therefore, the seemingly supermajoritarian levels of climate policy support reported in their Figure 2 are actually not indicative of support consistently exceeding 50% although their writing suggests otherwise. Yet, the claim that there exists majority support for unilateral climate policy that is void of reciprocal elements has been made repeatedly. For example, in “Unilateral or Reciprocal Climate Policy? Experimental Evidence from China” (Bernauer et al. 2016), a study that is of direct relevance to our research question, the authors conclude that “the results show that there is, perhaps surprisingly, strong and robust public support for unilateral, non-reciprocal climate policy” (Abstract) and that “we find that support for unilateral climate policy is rather strong and also robust, in the sense of not changing significantly even when participants are treated with information on positive or negative consequences of unilateral climate policy, or with negative news on GHG reduction policies of other key countries” (p. 161). We note that the latter sentence implies that climate policy support does not respond to information about policy impacts. This is also important to our study since we propose the effectiveness mechanism as one potential explanation for the appeal of multilateralism. We feel that at least for the subset of the literature that is most relevant to our research question, we should acknowledge these diverging findings. We have made revisions that reflect our updated, more nuanced perspective on the literature in track mode (p. 2 and 3).

2. *“There are a number of findings here that conventional wisdom models of climate action would not have predicted. For instance, this paper has a fascinating finding in its conjoint experiment showing that respondents are sensitive to cost structures in other countries (lines 185 to 193). The authors speculate on reasons for this (e.g. other-regarding preferences), and correctly use this as evidence that actions by other countries reshapes domestic political incentives to confront the climate crisis. However, classic models of free-riding would probably predict a different result: that as other countries take on more and more of the burden of confronting the climate crisis, support in the home country would go up, since this would provide more capacity for the home country to free-ride off the other country's costly actions. I may be thinking about this in too crude terms, but it would be good for the paper to walk the reader through these observable implications more explicitly.”*

We have improved our presentation and theoretical discussion of these results that now explicitly spell out R1's line of reasoning (p. 8, 3rd paragraph).

3. *“Reaffirming this same point, the article's claim (lines 27-30) that “the theory of the case is that voters will be more willing to contribute to the global public goods if other countries are doing so as well because multilateral policies are more likely to be effective, i.e., able to realize important social, economic, and environmental sustainability goals, and, under these agreements, resonate with reciprocal fairness norms.”...this seems like a valid conclusion from this article's empirical findings, but it does not clearly specify the priors most people working in this space might have. I can't think of any existing work on climate negotiations which arrives at this position as its core theory.”*

As discussed in our response to your first set of comments, we think the literature on the importance of multilateralism is more heterogeneous than suggested here. We have revised the manuscript to make this more explicit. We also think we provide new and useful empirical tests as we rely on different research strategies and novel data and hope that these results are also informative for scholars who share your prior. In terms of why multilateralism matters, we have revised the text to better explain our expectation which is based on two pillars. First, there is wide agreement that individuals value policies that provide them with benefits. In the context of climate action, these benefits arise from reaching social, economic, and environmental goals. Since climate change requires many countries to make significant investments in reducing emissions, the preference for multilateralism may originate from beliefs about the higher impact/effectiveness of such efforts. The initial idea behind this reasoning has already been proposed in Bechtel/Scheve (2013) but has not yet been evaluated in a design framework that would allow for a causal test. Second, individuals hold important fairness norms that affect their willingness to support the provision of public goods. A large literature (see, e.g., Fehr/Fischbacher 2002) has shown that one of the norms that are important to overcome public goods problems is reciprocity, i.e., the willingness to make contributions to realize a collective goal if others contribute as well. We have clarified this in the manuscript on p. 2.

4. *“In short, I think the article would benefit from a more direct effort to characterize its results in relation to classic accounts of global climate policy being shaped by free-riding and other simple models of climate negotiations as a simple collective action problem. (Notably, the word free-riding is not used once in the article!). I don't think serious scholars would argue that there is no interdependence or conditionality in global climate policies (and certainly none of the cites here make that suggestion). Instead, the value of this paper lies in shedding new light on the structure and dynamics of this interdependence. To the degree in which these forms of reciprocity are distinct from conventional game-theoretic accounts, this deserves to be surfaced. It may well be that the authors think the evidence here, on balance, supports conventional free-riding accounts of climate institutions, but this then deserves to be surfaced explicitly. This would be particularly helpful since the article provides some of the strongest evidence I have seen isolating the specific role of fairness norms in structuring multilateral incentives to pass climate policy, which is incredibly important to thinking about the conditions under which effective and equitable global climate policies are possible. This also includes a really important finding in the conjoint experiment - the result that interaction effects are largest when the other countries' average costs matches the respondents' country.”*

As discussed in the response to R1's first point, the goal of our paper is to estimate the effect of multilateralism on public support for climate action rather than to explicitly evaluate freeriding in global public goods problems. Where appropriate, we have added some discussion as requested by the reviewer of where our study speaks to public policy opinions that are or are not broadly consistent with freeriding

concerns in global public goods provision (p. 8, 3rd paragraph). Given the space constraints and to avoid over-selling our results given the goal of our study, we refrain from a more detailed discussion of freeriding in this manuscript.

Reviewer 2

1. *“A very interesting read on a relevant and pressing topic. The comments provided on this article are restricted to the field of my own specialisation, namely international law. I am therefore not able to assess the methodology in critical detail. As a lawyer, a standout question for me is how ‘multilateralism’ is concretely conceived in this study, and how it relates to reciprocity. It could be the case that voters look at other states’ policies, while the states themselves do not participate in any form of multilateralism / cooperation. Perhaps reference to the shared global temperature goal in the Paris Agreement may be of value for the conceptual framework.”*

We offer a description of how we conceptualize multilateralism on p. 2 (last paragraph) and have followed up on R2’s suggestion: “We study multilateralism in the context of climate cooperation where countries engage in “decentralized policy coordination” (20) which typically entails declaratory statements and agreements that specify policy objectives (e.g., the goal to limit global warming to well below 2°C as set out in the Paris agreement) and possible means, but lack enforcement mechanisms (21).”

2. *“More specific comments can be found attached as annotations in the article PDF.”*

For convenience and transparency we have listed these comments below. In addition, we have tracked all changes made in response to the reviewers’ comments in the manuscript file as requested by the editor. We also note that the line numbers that the reviewer used in her/his comments have changed due to the changes we made to the manuscript and due to having converted the manuscript into a word file to facilitate tracking edits. Our responses therefore also indicate where we have made a specific change (page and paragraph).

3. *Line 8: “Is there a difference between ‘opposition’ and ‘political opposition’?”*

In the context of our contribution the difference would be that political opposition comprises disapproval of policies meant to address climate change, whereas the more general term “opposition” could also entail economic or social responses such as the reluctance to buy energy-efficient appliances or reducing social activities that cause high CO₂ emissions (e.g., family trips that involve long-distance air travel). We have re-read the manuscript to ensure that the meaning of the term is clear when used.

4. *Line 15: “Is this erosion against climate action as a whole, or against the single (domestic) political choices on and how to distribute costs?”*

This sentence means that support for climate policy in general has been shown to decrease as costs increase. We have edited this sentence to make this clearer (p. 2, 2nd paragraph).

5. *Line 20: “It may be helpful for the reader if you specify concrete examples. For climate change, a central cooperative framework is of course the UNFCCC.”*

We have added the UNFCCC as the most prominent example (p. 2, 3rd paragraph).

6. *Line 29: “Perhaps worth spelling out: also, the concrete goal of limiting global temperature increase, which can only effectively be done collectively (i.e., to well below under 2 degrees Celsius as specified in the Paris Agreement)”*

We have added this clarification (p. 2, 3rd paragraph: “...to reach the collective goal of limiting global warming.”).

7. *Line 40: “Most climate cooperation lacks enforcement mechanisms I assume. Would this be a defining characteristic of (a subset of) multilateralism?”*

We agree with the reviewer. We explicitly note this in the paper: “We study multilateralism in the context of climate cooperation,..., which typically entails declaratory statements and agreements that specify policy objectives and possible means, but lack enforcement mechanisms” (p. 3, 1st paragraph).

8. *Line 52: “The definition / role of cooperation in the context of multilateralism would seem important here, as this phenomenon could manifest without actual multilateral cooperation between states. The UNFCCC framework is for example quite open, allowing states to determine their own policies on how to achieve their NDCs, without any concrete cooperation (in the sense of interaction) required (though options are foreseen, particularly in art 6 PA).”*
We agree and have added a sentence on p. 3, 2nd paragraph explicitly referring to the UNFCCC, the NDCs, and the point above.
9. *Line 66: “Perhaps helpful to indicate that this is domestic? (in light of EU's new envisioned carbon border adjustment mechanism which would capture foreign producers, possibly affecting voters' perception of the fairness of cost distribution).”*
We thank the reviewer for this suggestion which highlights that the level of decisionmaking may affect the willingness to back costly policies. We have clarified this in the text (p. 3, 4th paragraph).
10. *Line 97: “Is the assumption that these are linked, or coordinated in some way?”*
We do not make any assumptions about these being explicitly linked as we are interested in whether public support shifts already in response to being informed about the climate policy efforts of other countries in the absence of strongly institutionalized international interaction. We also think that this is a harder test for the multilateralism argument because formal international agreements may instill a sense of obligation that may be absent from the types of loose policy coordination most relevant in a world of "polycentric" and "weak" climate cooperation consistent with the flexibility offered by the UNFCCC (see also point 8 above). We have added a sentence on p. 3, 4th paragraph to clarify this.
11. *Line 149: “about? regarding?”*
This means that individuals have preferences over climate policies that other countries pursue.
12. *Line 168-171: “Very interesting and relevant.”*
Thank you.
13. *Line 185: “?”*
Thank you for noting this typo. We have reworded this sentence.
14. *Line 187: “Are these the only two? Is there a reason these two emerge?”*
There may be other explanations, but we believe the two we propose are theoretically relevant (see also our response to R1's point 1 on the significance of this finding for the ongoing debate over freeriding in climate cooperation).
15. *Line 188: “It would be helpful if you could explain why pressure is generated, particularly in light of the global public goods characteristics.”*
We have made revisions to better explain the origin of this pressure in the context of public good provision: individuals may have internalized a reciprocity norm that induces them to feel compelled to increase their own contribution to a public good as others contribute more, i.e., accept higher costs (p. 8, 3rd paragraph).
16. *Line 193: “This is interesting. Would you perhaps see a role for the multilaterally supported norm of 'common but differentiated responsibilities' here?”*
We believe that our results could be interpreted to provide a voter preference-based rationale for this UNFCCC principle. We have added a sentence to acknowledge this (p. 8, 4th paragraph).
17. *Line 209: “Are these multiple norms? Perhaps you could explain/ define further.”*
Yes, and we explore two types of reciprocity norms in this study (qualitative reciprocity and exact reciprocity).
18. *Line 211: “Here you mention agreements. This would be a helpful element/ criterion when defining multilateralism above.”*
The term "agreements" is now part of our working definition of climate multilateralism stated on p. 3, 1st paragraph.
19. *Line 229: “What is the relationship between multilateralism / cooperation, and reciprocity? Could they exist independently?”*

Multilateralism is a mode of international cooperation. Reciprocity in our context is a fairness norm that acts as a mechanism explaining the appeal of multilateralism understood as the willingness to contribute to the provision of a public good if others are also contributing (p. 11, 2nd paragraph).

20. Line 234: *“Interesting indeed.”*

Thank you.

21. Line 263: *“Perhaps specify to which results / factors this conclusion pertains.”* We have revised this sentence to clarify this (p. 11, 1st paragraph).

22. Line 270: *“Would this be analogically applicable to these quite different types of sustainability challenges? They are issues of global commons and public goods, yet not entirely as 'shared' perhaps as GHGs.”*

We share the reviewer's interest in this question and our last sentence (p. 12, 1st paragraph) is meant to identify this as a promising avenue for subsequent research.

Reviewer 3

1. *“My main concern lies in the implication and interpretation of the results. The authors contrast their study against a few recent publications and state (p.3): “these studies suggest that public support costly climate policies does not meaningfully depend on whether other countries are also contributing or not.” There are two things that I think need clarification. First, while the four papers cited differ in what they say and do, I don't think any of these say that public support is independent on what happens elsewhere. For instance, the Baiser-McGrath and Bernauer paper find that “citizens ... have strong preferences over the design of international climate agreements and contributions of other countries to the global effort” (from their abstract). Likewise, Aklin and Mildenerger focus on free-riding and mutual defection and don't claim that there is no interdependence at all. I'm not saying that this is a straw man, but I see more connections between this set of papers and this manuscript. For instance, the authors' own results on norms (Figure 2) are not inconsistent with these earlier findings. I'm not saying that they are perfectly consistent either, but I don't quite see them as contradicting each other.”*

We thank R3 for this comment which is similar to the R1's point #1. We agree that the existing literature deserves a nuanced discussion that also allows us to better position our own contribution. We have carefully re-read the literature on climate multilateralism to present a more nuanced portrayal of previous studies (although this summary admittedly has to remain quite brief given the space constraints). We agree that some studies offer evidence that seems explicitly or implicitly consistent with the public valuing international cooperation on climate change (e.g., Tvinneraum/Lachapelle 2016; Tingley/Tomz 2014; Bechtel/Scheve 2013).

Yet, several recent studies make explicit claims about the limited relevance or irrelevance of the behavior of other countries for our understanding of climate policy preferences. We deem it important that our review of this literature does justice to its results and how the authors have described their own findings even, or especially, if these diverge from our own results. For example, the study that R3 refers to (Beiser-McGrath/Bernauer 2019 NCC) states that “failure of other countries to act does not, per se, reduce the public's appetite for taking action against climate change. This is most evident in China, where the behaviour of other countries has effectively no impact on preferences regarding climate policy design. Whereas we find some effects of information on other countries' behaviour on preferences regarding particular attributes of climate policy design, this does not lead to a substantive decrease in overall support for the most popular international climate policy.” (p. 251). This is also how others have interpreted and prominently communicated these “unilateralism” results: “Writing in *Nature Climate Change*, Liam Beiser-McGrath and Thomas Bernauer show that publics in China and the United States

maintain their support for unilateral climate reforms, even when they learn that other countries are not acting” (Mildenberger 2020, p. 187).

To be fair, some of the key findings reported in Beiser-McGrath and Bernauer (2019 NCC) are based on an erroneous interpretation of their forced-choice conjoint data. This is because in this data, the baseline level of support for an agreement is 50% by design. Therefore, 50 percentage points of their predicted levels of support for different unilateral climate policy scenarios (Figure 2 in their study) are an artifact of their design and not indicative of supermajoritarian levels of approval (although their writing suggests otherwise). However, the claim that support for unilateral climate policy is void of reciprocal elements has also been made explicitly in other previous work. In “Unilateral or Reciprocal Climate Policy? Experimental Evidence from China” (Bernauer et al. 2016), a study that is of direct relevance to our research question, the authors conclude that “the results show that there is, perhaps surprisingly, strong and robust public support for unilateral, non-reciprocal climate policy” (Abstract) and that “we find that support for unilateral climate policy is rather strong and also robust, in the sense of not changing significantly even when participants are treated with information on positive or negative consequences of unilateral climate policy, or with negative news on GHG reduction policies of other key countries” (p. 161). We note that the latter sentence also implies that climate policy support does not respond to information about policy impacts, a finding that is important to our study since we propose an effectiveness mechanism as one potential explanation for the appeal of multilateralism. Although we share R3’s wish to present a summary of the existing literature that would suggest more consistency in its key findings, we feel that at least for the subset of the literature that is most relevant to our research question, we should acknowledge these diverging findings. We have made revisions that reflect our updated, more nuanced perspective on the literature in track mode (p. 2 and p. 3).

2. *“Second, let’s assume that the authors disagree with my previous point and insist on contrasting multilateralism against unconditional action (fair enough!). I then don’t see their results as particularly supportive of the multilateralism hypothesis. Focusing on the first set of the results, we find that 53% are unconditionally supportive and this number only goes up by a mere 6% when multilateralism is introduced. The authors argue that the 53% “is merely a slim majority that can be broken by relatively small shifts in public support.” But what’s the size of those who are conditional cooperators? Based on these results, it seems that it’s a small groups (certainly smaller than the 53% by quite a margin. This is a long and convoluted way to say: there is a difference between saying that multilateralism helps increase support (which the authors show convincingly) and that it reflects the way a large segment of the population thinks (where it seems to me that the authors show the opposite).”*

This comment has led us to think more thoroughly about the level of support and the magnitude and relevance of the multilateralism effect. Clearly, we agree that only a subset of the population may be conditionally cooperative. Moreover, our main point is indeed to say that multilateralism helps increase support without overstating its magnitude. We have revised the paper to give a balanced assessment of the size of the multilateralism effect. To help bolster our view that it is important (and that importance is not necessarily equivalent to effect size), we think it is helpful to think about the core concern of generating majority support (or even supermajoritarian levels of support) even if the sources of this support will vary. In addition, we can assess the magnitude of this effect in several ways that go beyond its statistical significance. A common approach is to benchmark the effect against the baseline level of policy approval observed in the control condition. If we adopt this approach, our treatment effect is equivalent to an 11% increase over the baseline level of carbon tax approval. We think this is politically important but agree with R3 that we need to not exaggerate its magnitude (see revisions on p. 4). More generally, we think inducing an opinion change of this magnitude is typically viewed as substantively significant in survey experiments across different issue domains. Finally, since we are exploring a

policy-relevant question, we may also want to assess the effect in terms of its potential to bring about decisive shifts in majority support. As we report in Figure 1, average levels of support are 53% and this estimate is not significantly different from 50%. Therefore, in the absence of multilateral policy coordination, we would not even be certain that a majority supports the introduction of a carbon tax. In contrast, a multilateral approach raises carbon tax approval to 59% and this estimate is significantly greater than 50%. Therefore, we document an instance where multilateralism may be able to bring about a pivotal shift by creating majority support for a costly climate policy. We have made edits in the manuscript (again, in track mode) that continue to view the magnitude of the effect as politically important but without exaggerating its size (p. 4).

3. *“I have two questions on the conjoint analysis.*

a. *First, I would like to hear more from the authors about the results regarding the cost level for other countries (Figure 3). If I read this correctly, respondents became less supportive the more others had to pay. I'm trying to think about how to square this with the idea what we generally prefer if others pay more. Is this finding telling us about how people evaluate these costs? Or, alternatively, did readers misunderstand the prompt? (More generally, it wasn't clear to me whether the authors included comprehension checks or not.)”*

We agree with R3's expectation that in a standard public goods contribution framework one would expect learning that someone else, another country, is contributing more should make respondents more, not less, supportive. The public good is getting paid for by someone else, all else equal. We now recognize this clearly and then provide two potential (and complementary) explanations for our findings in the opposite direction on p. 8 and 9. First, respondents may want to keep climate costs low in other countries due to altruistic inclinations. Second, respondents may have been concerned that any approach to the problem that was more costly in another country could eventually increase costs in their own country due to norms of reciprocity. At the same time, new research would have to be carried out to fully evaluate these arguments.

*“Second, in appendix (p.1), the authors say that they interviewed 2,000 respondents per country (except in the US, where they interviewed 4,081 people). They also say that they repeated the conjoint 8 times for each respondent. But Figure A6 says that the sample size is 32,000 for France, Germany, and the UK. Shouldn't it be half (2,000*8 tasks)? Am I missing something?”*

There are 2,000 respondents in each of the three European countries and each respondent assessed two scenarios per round (of which there were 8). Multiplying these numbers we obtain 32,000 which is the number of observations in each of the European countries.

4. *“p.5 ‘To explore the role of effectiveness in the eliminated effects framework...’ - I found the wording of that paragraph a bit opaque and hard to follow (even though I'm familiar with the Acharya et al piece!). Maybe it's worth editing it for readers who are not familiar with this approach (or expand in the methods section)? Maybe a figure to clarify what is being compared to what? I don't have a good suggestion, but given that these are important results, it would be good to have a clear explanation.”*
We have simplified the language here to make the design easier for readers to understand (p.4, 3rd paragraph).

REVIEWERS' COMMENTS

Reviewer #1 (Remarks to the Author):

I thought this was a terrific paper upon my first review, and having re-read the article, I think the authors have helpfully contextualized their theory and empirics in ways that fully satisfy my comments. I do take their point that some literatures have been more stark in their claims about unilateralism than I suggested in my previous review, and think they position the paper well in this context. They have taken on board some of the reviewers' suggestions, but I am also persuaded by the elements they have chosen to disagree with.

I strongly recommend the piece be published, and think it will become an important contribution to this broader debate over conditionality in global climate policymaking.

Reviewer #2 (Remarks to the Author):

The authors helpfully explain their revisions which adequately address the comments provided on the first version.

I have two very minor language/ definitional points from the perspective of multilateralism under public international law. These pertain to page 3.

1. p. 3, line 104: In their response to the reviewers, the authors helpfully clarify that they are not focusing on 'linked' carbon taxes (or 'clubs'). I would suggest spelling this out. The phrase 'implemented by... multiple countries' can be interpreted as referring to a linked tax / international cooperation between tax schemes. As a linked tax is on the policy table, this may cause confusion for certain readers.

2. p.3. line 108: It can be somewhat confusing to refer to 'NDCs to reducing global emissions' as a 'principle', given the nebulous meaning of this concept under the public international law framework in which the UNFCCC resides. NDCs would in any event not constitute a 'principle'. Suggestion: NDC 'mechanism' or '(bottom-up)-approach'.

Reviewer #3 (Remarks to the Author):

Thank you for engaging with my comments so thoroughly. And good catch on the Beiser-McGrath and Bernauer piece. I hadn't read it closely, but I probably would have missed it.

Revision Memo

Response to the Reviewers

Reviewer #1 (Remarks to the Author):

"I thought this was a terrific paper upon my first review, and having re-read the article, I think the authors have helpfully contextualized their theory and empirics in ways that fully satisfy my comments. I do take their point that some literatures have been more stark in their claims about unilateralism than I suggested in my previous review, and think they position the paper well in this context. They have taken on board some of the reviewers' suggestions, but I am also persuaded by the elements they have chosen to disagree with. I strongly recommend the piece be published, and think it will become an important contribution to this broader debate over conditionality in global climate policymaking."

We thank the reviewer for these encouraging words and the constructive feedback on our manuscript.

Reviewer #2 (Remarks to the Author):

"The authors helpfully explain their revisions which adequately address the comments provided on the first version."

We thank the reviewer for the constructive feedback on our manuscript.

"I have two very minor language/ definitional points from the perspective of multilateralism under public international law. These pertain to page 3.

1. p. 3, line 104: In their response to the reviewers, the authors helpfully clarify that they are not focusing on 'linked' carbon taxes (or 'clubs'). I would suggest spelling this out. The phrase 'implemented by... multiple countries' can be interpreted as referring to a linked tax / international cooperation between tax schemes. As a linked tax is on the policy table, this may cause confusion for certain readers."

We have added a sentence highlighting that we do not study the more institutionalized approach of creating "linked carbon taxes".

2. p.3. line 108: It can be somewhat confusing to refer to 'NDCs to reducing global emissions' as a 'principle', given the nebulous meaning of this concept under the public international law framework in which the UNFCCC resides. NDCs would in any event not constitute a 'principle'. Suggestion: NDC 'mechanism' or '(bottom-up)-approach'.

We have deleted the word "principle".

Reviewer #3 (Remarks to the Author):

"Thank you for engaging with my comments so thoroughly. And good catch on the Beiser-McGrath and Bernauer piece. I hadn't read it closely, but I probably would have missed it."

We thank the reviewer for the constructive feedback on our manuscript.